# Density-dependent resistance protects *Legionella pneumophila* from its own antimicrobial metabolite, HGA

Tera C Levin[1]*, Brian P Goldspiel[1†], Harmit S Malik[1,2]

[1]Division of Basic Sciences, Fred Hutchinson Cancer Research Center, Seattle, United States; [2]Howard Hughes Medical Institute, Fred Hutchinson Cancer Research Center, Seattle, United States

**Abstract** To persist in microbial communities, the bacterial pathogen *Legionella pneumophila* must withstand competition from neighboring bacteria. Here, we find that *L. pneumophila* can antagonize the growth of other *Legionella* species using a secreted inhibitor: HGA (homogentisic acid). Unexpectedly, *L. pneumophila* can itself be inhibited by HGA secreted from neighboring, isogenic strains. Our genetic approaches further identify *lpg1681* as a gene that modulates *L. pneumophila* susceptibility to HGA. We find that *L. pneumophila* sensitivity to HGA is density-dependent and cell intrinsic. Resistance is not mediated by the stringent response nor the previously described *Legionella* quorum-sensing pathway. Instead, *L. pneumophila* cells secrete HGA only when they are conditionally HGA-resistant, which allows these bacteria to produce a potentially self-toxic molecule while restricting the opportunity for self-harm. We propose that established *Legionella* communities may deploy molecules such as HGA as an unusual public good that can protect against invasion by low-density competitors.
DOI: https://doi.org/10.7554/eLife.46086.001

*For correspondence:
tlevin@fredhutch.org

Present address: †Perelman School of Medicine, University of Pennsylvania, Philadelphia, United States

Competing interests: The authors declare that no competing interests exist.

## Introduction

Inter-bacterial conflict is ubiquitous in nature, particularly in the dense and highly competitive micro-environments of biofilms (*Davey and O'toole, 2000*; *Foster and Bell, 2012*; *Rendueles and Ghigo, 2015*). In these settings, bacteria must battle for space and nutrients while evading antagonism by neighboring cells. One strategy for managing these environments is for bacteria to cooperate with their kin cells, sharing secreted molecules as public goods (*Nadell et al., 2016*; *Abisado et al., 2018*). However, these public goods are vulnerable to exploitation by other species or by 'cheater' bacterial strains that benefit from public goods but do not contribute to their production. For this reason, many bacteria participate in both cooperative and antagonistic behaviors to survive in multi-species biofilms. Bacterial antagonistic factors can range from small molecules to large proteins, delivered directly or by diffusion, and can either act on a broad spectrum of bacterial taxa or narrowly target only a few species. Although narrowly targeted mechanisms may seem to be of less utility than those that enable antagonism against diverse bacterial competitors, targeted strategies can be critical for bacterial success because they tend to mediate competition between closely-related organisms that are most likely to overlap in their requirements for restricted nutrients and niches (*Hibbing et al., 2010*).

The bacterium *Legionella pneumophila* (*Lp*) naturally inhabits nutrient-poor aquatic environments where it undergoes a bi-phasic lifestyle, alternating between replication in host eukaryotes and residence in multi-species biofilms (*Lau and Ashbolt, 2009*; *Declerck et al., 2007*; *Declerck, 2010*; *Taylor et al., 2013*). If *Lp* undergoes this lifecycle within man-made structures such as cooling towers, the bacteria can become aerosolized and cause outbreaks of a severe, pneumonia-like disease

**eLife digest** In the environment, bacteria frequently compete with each other for resources and space. These battles often involve the bacteria releasing toxins, antibiotics or other molecules that make it more difficult for their neighbors to grow. The bacteria also carry specific resistance genes that protect them from the effects of the molecules that they produce.

*Legionella pneumophila* is a species of bacteria that infects people and causes a severe form of pneumonia known as Legionnaires' disease. The bacteria spread in droplets of water from contaminated water systems such as sink faucets, cooling towers, water tanks, and other plumbing systems. In these water systems, *L. pneumophila* cells live within communities known as biofilms, which contain many different species of bacteria. These communities often include other species of *Legionella* that compete with *L. pneumophila* for similar nutrients. However, *L. pneumophila* was not known to produce any toxins or antibiotics, so it was not clear how it is able to survive in biofilms.

Levin et al. used genetic approaches to investigate how *L. pneumophila* competes with other species of *Legionella*. The experiments found that this bacterium released a molecule called homogentisic acid (HGA) that reduced the growth of neighboring *Legionella* bacteria. Unexpectedly, *L. pneumophila* was not always resistant to HGA, despite secreting large quantities of this molecule. Instead, *L. pneumophila* cells were only resistant to HGA when the bacteria were living in crowded conditions.

Previous studies have shown that HGA is widely produced by bacteria and other organisms – including humans – but this is the first time it has been shown that this molecule limits the ability of bacteria to grow. The work of Levin et al. suggests that HGA may help *L. pneumophila* bacteria to persist in biofilms, but more work needs to be done to test this idea. A possible next step is to test whether drugs that inhibit the production of HGA can eliminate *Legionella* bacteria from water systems. If so, similar treatments could potentially be used to stop and prevent outbreaks of Legionnaires' disease in the future.

DOI: https://doi.org/10.7554/eLife.46086.002

in humans, called Legionnaires' disease (*Fraser et al., 1977*; *McDade et al., 1977*; *Fields et al., 2002*). Because of the serious consequences of *Lp* colonization, the persistence and growth of *Legionella* in aquatic environments has been the subject of numerous studies. These studies have examined replication within protozoan hosts (*Rowbotham, 1980*; *Lau and Ashbolt, 2009*; *Hoffmann et al., 2014*), survival in water under nutrient stress (*Li et al., 2015*; *Mendis et al., 2015*), and sensitivity to biocides (*Kim et al., 2002*; *Lin et al., 2011*). Here, we focus on interbacterial competition as an underappreciated survival challenge for *Lp*.

*Legionella spp.* are not known to produce any antibiotics, bacteriocins, or other antibacterial toxins. Bioinformatic surveys of *Legionella* genomes have revealed a number of polyketide synthases and other loci that likely produce bioactive metabolites (*Johnston et al., 2016*; *Tobias et al., 2016*), but these have not been shown to exhibit any antimicrobial functions. Nevertheless, there are some hints that interbacterial competition is relevant for *Lp* success within biofilms. For example, one study of artificial two-species biofilms found that viable *Lp* were able to persist for over two weeks in the presence of some bacterial species (e.g. *Pseudomonas fluorescens*, *Klebsiella pneumoniae*) but not others (e.g. *Pseudomonas aeruginosa*) (*Stewart et al., 2012*). Additionally, *Lp* bacteria are often co-resident with other *Legionella spp.* in man-made structures, with some studies showing that *Lp* proliferation is correlated with a decrease in other *Legionella spp.* populations (*Wéry et al., 2008*; *Pereira et al., 2017*; *Declerck et al., 2007*). These studies suggest that *Lp* bacteria may compete with other *Legionella spp.* for similar biological niches.

The most direct evidence for interbacterial competition comes from *Stewart et al. (2011)*, who found that *Lp* could antagonize the growth of neighboring *Legionella spp.* on the same plate. The molecules mediating this competition have not been identified, although previous work suggested a role for *Lp*'s secreted surfactant, a thin liquid film that facilitates the spread of *Lp* across agar plates (*Stewart et al., 2009*; *Stewart et al., 2011*). Still, it remained unknown if surfactant played a direct or indirect role in inter-*Legionella* inhibition.

Here, we use unbiased genetic approaches to find that homogentisic acid (HGA) produced by *Lp* generates oxidative intermediates that inhibit the growth of neighboring *Legionella spp.* We find that HGA production co-occurs with surfactant production, but that these are independent, separable phenomena. The redox state of HGA appears to be critical for its activity, as HGA is only toxic in aerobic conditions and fully oxidized HGA-melanin pigment is inactive. Unexpectedly, we find that although *Lp* secretes abundant HGA, it is also susceptible to HGA-mediated inhibition. We identify one gene– *lpg1681*– that enhances *Lp* susceptibility to HGA. Moreover, we find that *Lp* cells are resistant to HGA at high-density, which is also when they secrete large amounts of HGA. This high-density resistance is cell intrinsic and is independent of growth phase, the stringent response, or the previously described quorum-sensing pathway in *Legionella*. Based on these findings, we propose that HGA has the potential to play an important role in structuring *Legionella* communities.

## Results

### *L. pneumophila* inhibits *L. micdadei* via an unknown, secreted inhibitor

Inspired by previous reports (*Stewart et al., 2011*), we investigated how *Legionella pneumophila* (*Lp*) engages in inter-*Legionella* competition. We found that *Lp* inhibited the growth of neighboring *Legionella micdadei* (*Lm*) plated 1 cm away on solid media, suggesting that it produces a secreted inhibitor (*Figure 1A*). This inhibition was most robust when we plated the *Lp* strain on low-cysteine media 3–4 days prior to plating *Lm*, allowing time for the inhibitory molecule to be produced and spread across the plate. To quantify this inhibition, we recovered *Lm* grown at different distances from *Lp*. After 48 hr incubation, we found a 10,000-fold difference in growth between *Lm* antagonized by *Lp* versus *Lm* plated outside of the zone of inhibition (*Figure 1B*).

Previous studies (*Stewart et al., 2011*) had proposed that inter-*Legionella* inhibition could be caused by *Lp*'s secreted surfactant, which is produced by *Lp* but not *Lm* (*Stewart et al., 2009*). We tested this hypothesis by deleting a surfactant biosynthesis gene, *bbcB*, from the *Lp* genome (*Stewart et al., 2011*). The resulting Δ*bbcB* strain did not produce surfactant (*Figure 1—figure supplement 1B*), yet it still inhibited adjacent *Lm* (*Figure 1A*, *Figure 1—figure supplement 1C*). When quantified, we observed nearly identical inhibition from both wild type *Lp* and Δ*bbcB Lp* (*Figure 1B*

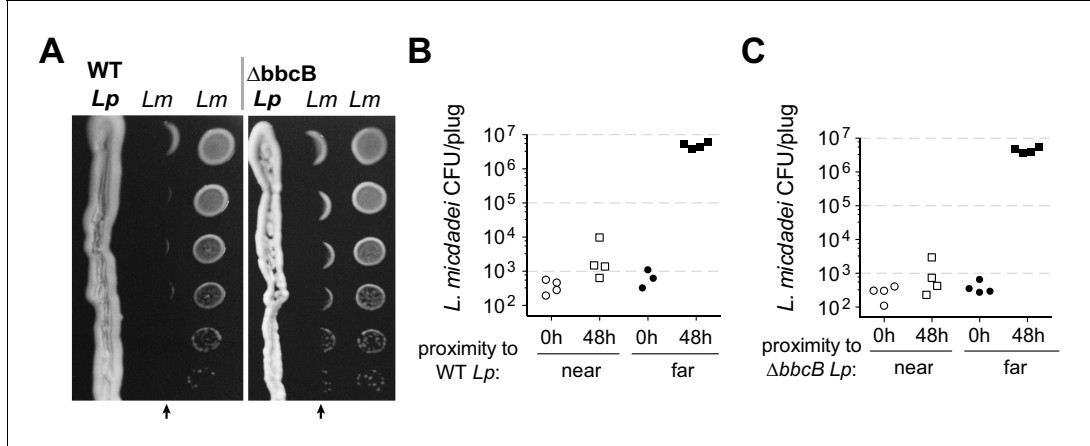

**Figure 1.** *L. pneumophila (Lp)* produces a secreted inhibitor independent of surfactant. (**A**) When pre-incubated on low-cysteine BCYE charcoal agar plates, *Lp* produces a zone of inhibition, impacting the growth of nearby *L. micdadei (Lm)*. Arrows mark the edge of inhibition fronts. Droplets of *Lm* at different dilutions were added to the plate in parallel columns three days after streaking *Lp*. WT *Lp* (left panel) generates a similar zone of inhibition as a surfactant-null mutant, Δ*bbcB* (right panel). (**B**) Quanitification of *Lm* growth within ('near') or outside of ('far') the wild type *Lp* zone of inhibition. (**C**) Quanitification of *Lm* growth within or outside of the Δ*bbcB Lp* zone of inhibition. In B and C, bacteria were sampled and removed from the plate in a 'plug' of fixed area before plating for viable CFUs.

DOI: https://doi.org/10.7554/eLife.46086.003

The following figure supplement is available for figure 1:

**Figure supplement 1.** Separation of surfactant and antimicrobial phenotypes.

DOI: https://doi.org/10.7554/eLife.46086.004

*and C*) indicating that the surfactant did not enhance inhibition. Furthermore, we observed that the zone of inhibition surrounding wildtype *Lp* did not always co-occur with the spread of the surfactant front (*Figure 1—figure supplement 1A*). From these results, we conclude that *L. pneumophila* can cause strong growth inhibition of neighboring *Legionella* using an unknown molecule that is distinct from surfactant.

## Transposon screen pinpoints HGA-melanin pathway in inter-*Legionella* inhibition

To determine which molecule(s) might be responsible for inter-*Legionella* inhibition, we performed an unbiased genetic screen in *Lp*. We generated *Lp* mutants using a drug-marked Mariner transposon that randomly and efficiently integrates into the *Legionella* genome (*O'Connor et al., 2011*). To identify mutants that were defective in producing the inhibitor, we transferred each mutant onto a lawn of *L. micdadei* on low-cysteine plates and examined the resulting zone of inhibition surrounding each *Lp* mutant (*Figure 2A*). After screening 2870 clones, we isolated 19 mutants that produced a smaller zone of inhibition than wild type *Lp*, as well as five mutants that showed a complete loss of inhibition (*Figure 2B*, *Supplementary file 1*). We refer to these as 'small zone' and 'no zone'

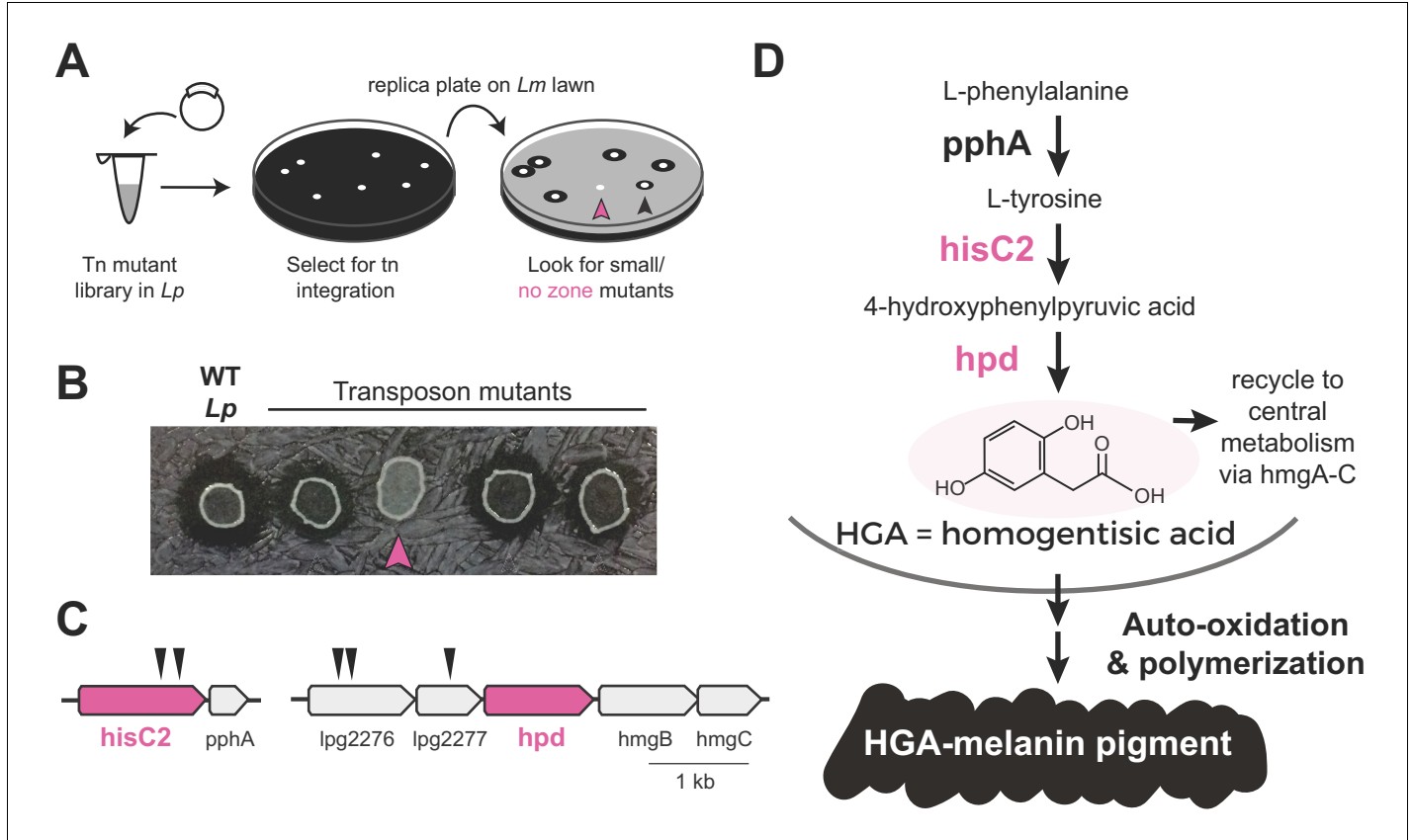

**Figure 2.** Transposon mutagenesis screen implicates the HGA-melanin pathway in production of the inhibitor. (**A**) Screen for mutant *L. pneumophila* (*Lp*) that do not inhibit *L. micdadei* (*Lm*). Following electroporation of a Mariner-transposon-containing plasmid, *Lp* mutants were selected for transposon integration. Colonies were patched or replica plated onto a lawn of *Lm*. Mutants of interest generated a zone of inhibition that was reduced (black arrowhead) or absent (pink arrowhead) compared to WT *Lp*. (**B**) Selected transposon mutants produce abnormal zones of inhibition when grown on a lawn of *Lm*. Pink arrowhead indicates a *hisC2*::Tn 'no zone' mutant. (**C**) Transposon insertion sites (triangles) identified in the five recovered 'no zone' mutants. (**D**) HGA-melanin synthesis pathway. HGA is exported from the cell where it auto-oxidizes and polymerizes to form HGA-melanin. Genes in pink were validated by complementation to have essential roles in *Lm* inhibition.

DOI: https://doi.org/10.7554/eLife.46086.005

The following figure supplement is available for figure 2:

**Figure supplement 1.** Genetic validation linking inhibition-defective mutants to the HGA-melanin pathway.
DOI: https://doi.org/10.7554/eLife.46086.006

mutants, respectively. Among the 'small zone' mutants, some had defects in surfactant spreading on plates, while others showed enhanced surfactant spread (*Figure 2—figure supplement 1A*), further distinguishing inter-bacterial inhibition from surfactant secretion.

We focused on the 'no zone' mutants, as these had the strongest defects in inhibition. These five mutants carried transposon insertions in two separate operons (*Figure 2C*). The first operon had two insertions in the *hisC2* gene (*lpg1998)*, which breaks down tyrosine as part of the HGA-melanin metabolic pathway (*Figure 2D*). Its downstream gene, *pphA*, converts phenylalanine to tyrosine in the same pathway. To validate the role of *hisC2* in inhibition, we overexpressed this gene in the *hisC2* transposon mutant background and found that *hisC2* alone was sufficient to complement the mutant phenotype (*Figure 2—figure supplement 1B*). Having confirmed the role of *hisC2*, we turned to the second operon, where we had recovered transposon insertions in two uncharacterized genes, *lpg2276* and *lpg2277* (*Figure 2C*). These two genes lie immediately upstream of *hpd* (*lpg2278*), which is known to act with *hisC2* in the HGA-melanin pathway (*Steinert et al., 2001*; *Gu et al., 1998*) (*Figure 2D*). Because transposon insertions at the beginning of an operon can disrupt the expression of downstream genes via polar effects, we hypothesized that the insertions we recovered in *lpg2276* and *lpg2277* altered inter-*Legionella* inhibition via disruption of *hpd* expression. Indeed, we were able to complement insertions in both genes, which had yielded 'no zone' mutants, by overexpressing *hpd,* despite the fact that *hpd* overexpression caused a growth defect (*Figure 2—figure supplement 1B*). In conclusion, all five 'no zone' isolates had mutations that disrupted the same metabolic pathway involved in the production of HGA-melanin. Consistent with these findings, we observed defects in HGA-melanin pigmentation in all of the 'no zone' mutants as well as some of the 'small zone' mutants (*Figure 2—figure supplement 1E*).

The HGA-melanin pathway is found in diverse eukaryotes and bacteria (*Nosanchuk and Casadevall, 2003*; *Liu and Nizet, 2009*) including *Legionella spp.* (*Fang et al., 1989*) (*Figure 2—figure supplement 1D*). This pathway produces homogentisic acid (HGA) from the catabolism of phenylalanine or tyrosine (*Steinert et al., 2001*) (*Figure 2D*). HGA can either be further metabolized and recycled within the cell via HmgA-C, or it can be secreted outside of the cell, where it auto-oxidizes and polymerizes to form a black-brown pigment called HGA-melanin, or pyomelanin (*Kotob et al., 1995*) (*Figure 2D*). To test whether intracellular metabolites downstream of HGA are necessary for inhibition, we deleted *hmgA*, the first gene in the pathway that recycles HGA back into central metabolism. We found that the Δ*hmgA* strain produced a zone of inhibition that was similar or slightly larger than wild type (*Figure 2—figure supplement 1C*). We therefore inferred that synthesis of secreted HGA and/or HGA-melanin, but not its recycling and intracellular processing, is required for *Lp* inhibition of *Lm.*

## HGA inhibits the growth of *Legionella micdadei*, but HGA-melanin does not

To our knowledge, the HGA-melanin pathway has not previously been implicated in inter-bacterial competition. To the contrary, prior work has emphasized the beneficial (rather than detrimental) effects of HGA-melanin on *Legionella* growth, by providing improved iron scavenging (*Chatfield and Cianciotto, 2007*) and protection from light (*Steinert et al., 1995*). We therefore asked whether the active inhibitor produced by the pathway was HGA-melanin, or alternatively a precursor molecule (*Figure 3A*). We tested the potential inhibitory activity of HGA-melanin pigment from *Lp* conditioned media in multiple experiments; however, we never observed any inhibition of *Lm*. We wished to rule out the possibility that the pigment secreted into rich media was too dilute to be active, or that other nutrients in the media might interfere with inhibition. We therefore isolated a crude extract of HGA-melanin from *Lp* conditioned media via acid precipitation (as in *Chatfield and Cianciotto, 2007*), washed and concentrated the pigment approximately 10-fold and repeated the assay; the concentrated pigment also showed no inhibitory activity (*Figure 3B and E*).

The first metabolite secreted by the HGA-melanin pathway is HGA. We tested whether HGA could behave as an inhibitor even though HGA-melanin could not. Indeed, we found that synthetic HGA robustly inhibited *Lm* growth, both when spotted onto a lawn of *Lm* and when titrated into AYE rich media (*Figure 3B and C*), We found that inhibition of *Lm* by HGA is relatively specific at the molecular level; neither 2-hydroxyphenylacetic acid nor 3-hydroxyphenylacetic acid, two HGA-related molecules that differ from HGA by only a single -OH group, were able to inhibit *Lm* growth at any concentration tested (*Figure 3—figure supplement 1A*).

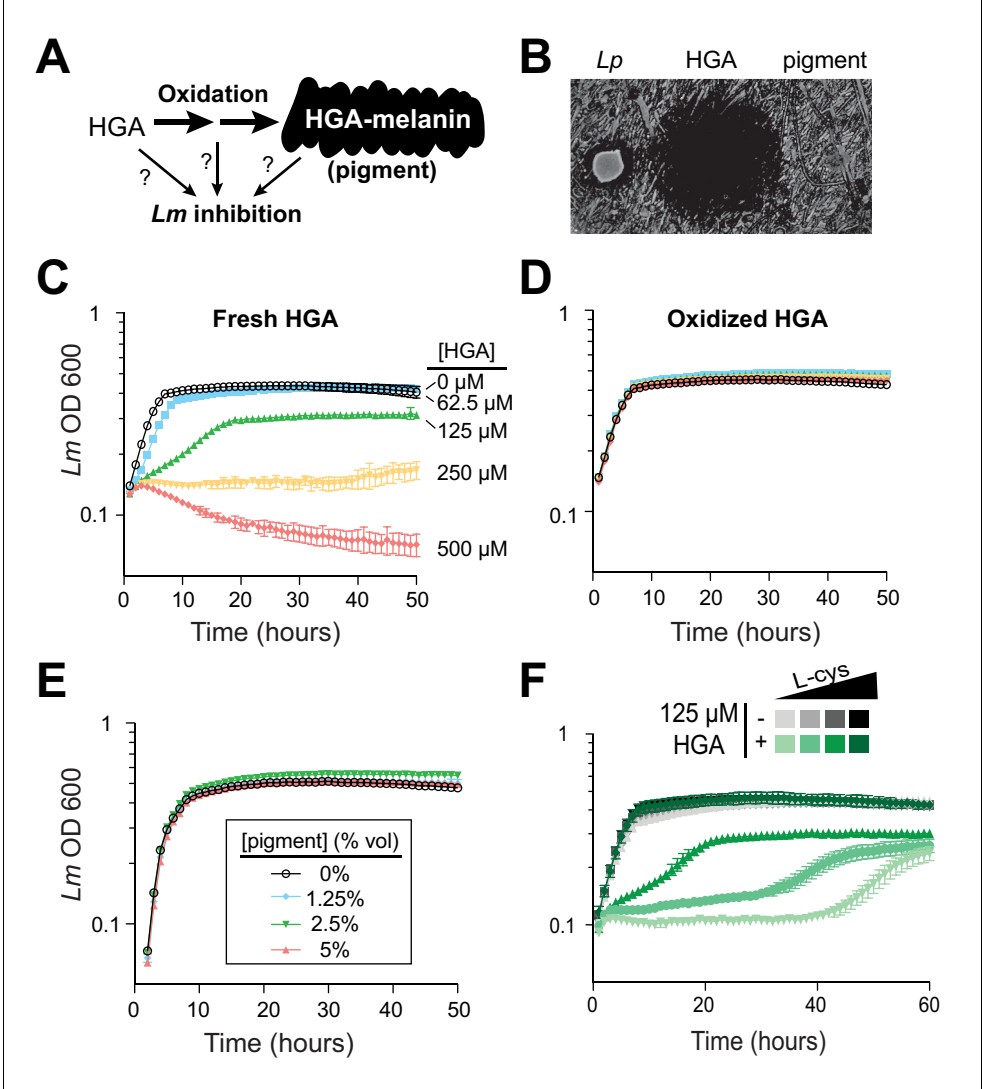

**Figure 3.** Synthetic HGA inhibits *Legionella micdadei* growth, depending on its oxidation state. (**A**) We tested whether inter-bacterial inhibition is caused by HGA, HGA-melanin, or an oxidative intermediate. (**B**) Zones of inhibition on a lawn of *Lm* generated from either live *Lp* bacteria, synthetic 50 mM HGA, or concentrated pigment extract. HGA prevents *Lm* growth in a large region (central dark circle) but pigment does not. (**C**) Growth inhibition of *Lm* from increasing concentrations of synthetic HGA in rich AYE media. (**D**) Pre-oxidation of synthetic HGA in AYE media for 24 hr eliminates its inhibitory activity, resulting in normal *Lm* growth. Concentrations of HGA colored as in panel C. (**E**) Addition of oxidized HGA-melanin pigment from *Lp* has little impact on *Lm* growth in AYE liquid media. (**F**) In the absence of HGA (gray symbols), titration of L-cysteine (L-cys) from 25–200% of standard AYE media has little impact on *Lm* growth. In contrast, HGA activity is enhanced in low-cysteine media and decreased in high-cysteine media (green symbols). All error bars show standard deviations among 3–4 replicates.

DOI: https://doi.org/10.7554/eLife.46086.007

The following figure supplement is available for figure 3:

**Figure supplement 1.** Impacts of chemical compounds on HGA-mediated inhibition of *Legionella*.

DOI: https://doi.org/10.7554/eLife.46086.008

Because HGA, but not HGA-melanin, can inhibit *Lm* growth (*Figure 3B and E*), we inferred that the oxidative state of HGA might be important to its inhibitory activity. HGA is a reactive molecule, which auto-oxidizes (*Eslami et al., 2013*) and polymerizes to form HGA-melanin through a series of non-enzymatic steps that are not genetically encoded (*Steinert et al., 2001*) and are therefore

undetectable by our genetic screen. Given its auto-oxidative potential, we next tested whether HGA might cause growth inhibition by oxidizing other nearby molecules, either in the media or on bacterial cells. We allowed synthetic HGA to oxidize completely for 24 hr in AYE media before adding *Lm* (*Figure 4—figure supplement 1D*). We found that pre-oxidation completely abolished synthetic HGA activity, even at very high HGA concentrations (*Figure 3D*, compare to 3C). This experiment also ruled out the possibility that HGA acts by causing nutrient depletion or other modifications of the media, since media pre-incubated with HGA for 24 hr is still able to support normal *Lm* growth. Instead, we infer that *Lm* inhibition results from direct interactions between bacterial cells and either HGA itself or unstable, reactive intermediates produced during HGA oxidation.

Small, reactive, quinone-like molecules similar to HGA are known to react with oxygen to produce $H_2O_2$, which is broadly toxic to bacteria (*Hassan and Fridovich, 1980*). In such cases, extracellular catalase has been shown to protect bacteria against the toxic effects of $H_2O_2$ (*Hassan and Fridovich, 1980*; *Imlay, 2013*). To test if HGA toxicity occurs via a similar mechanism as $H_2O_2$, we asked if extracellular catalase was sufficient to protect *Lm* from HGA-mediated toxicity. Even at very high catalase concentrations, we found that catalase provided no protection from HGA (*Figure 3—figure supplement 1B*), ruling out the production of extracellular $H_2O_2$ as a potential mechanism of action for HGA-mediated inhibition. We also considered the possibility that HGA as a weak acid could inhibit *Lm* indirectly by altering the local pH, but we observed that adding HGA at 1 mM into AYE media or PBS caused little to no change in pH.

Given that the redox state of HGA is critical for inhibition, we reasoned that it should be possible to modulate HGA activity by altering the redox state of the media using reducing agents. We accomplished this by titrating L-cysteine from 25% to 200% of the levels in standard AYE media. In the absence of HGA, these altered cysteine concentrations had little impact on *Lm* growth (*Figure 3F*, gray symbols). However, lower cysteine concentrations greatly sensitized *Lm* to HGA, while excess cysteine was completely protective (*Figure 3F*). These findings may help partially explain why HGA's inhibitory activity on *Legionella* has not been previously detected, as *Legionella* species are typically studied in cysteine-rich media. We found that synthetic HGA is readily able to react with cysteine in vitro (*Figure 3—figure supplement 1D*), presumably impacting the oxidation state of HGA. Moreover, incubation of HGA with two other reducing agents, DTT (dithiothreitol) or reduced glutathione, similarly quenched HGA's inhibitory activity (*Figure 3—figure supplement 1E*). From these experiments, we conclude that HGA is less potent in rich media because it reacts with excess cysteine (or other bystander molecules) before it can interact with *Lm*. In sum, these results implicate the reactive activity of HGA and/or its transient, oxidative intermediates in inter-*Legionella* inhibition.

In these experiments, synthetic HGA was a robust inhibitor of *Lm.* However, this inhibition required relatively high concentrations of HGA (>50 µM). We next quantified the amount of HGA secreted by *Lp* to determine if these levels were biologically relevant. Compared to other *Legionella spp.*, *Lp* produces much more pigment (*Figure 2—figure supplement 1D*), suggesting that it secretes considerably more HGA. To estimate the quantity of secreted HGA, we created a standard curve of synthetic HGA in AYE rich media. We allowed the HGA added to completely oxidize to HGA-melanin, which can be measured by OD 400 readings. In this way, we can use the pigment levels after oxidation as a reliable measure of total HGA that was produced by a given time-point (*Figure 4—figure supplement 1D*). Using this calibration, we estimated that wild type *Lp* had secreted the equivalent of 1.7 mM HGA after 48 hr of culture, whereas the hyperpigmented Δ*hmgA* strain secreted about 2.6 mM HGA. Thus, the levels of HGA produced by *Lp* are considerably higher than the inhibitory concentrations of synthetic HGA used in our assays (50–500 µM), at least under lab conditions. In contrast, we did not detect any pigment from the non-inhibitory *hisC2*::Tn strain (*Figure 4—figure supplement 1E*). From these experiments, we conclude that HGA is an abundant, secreted, redox-active metabolite of *Lp*, which can accumulate in concentrations that are relevant for inter-*Legionella* inhibition.

## *L. pneumophila* can be susceptible to its own inhibitor

Our results so far indicated that HGA can be a potent, redox-active inhibitor of *Lm*, which is volatile and capable of reacting with many types of thiol-containing molecules. If *Lp* uses HGA to compete with neighboring *Legionella spp.*, we anticipated that *Lp* would have evolved some form of resistance to its own secreted inhibitor. Therefore, we next tested *Lp* susceptibility to inhibition in low-

cysteine conditions, as we had previously done for *Lm*. Surprisingly, we found that *Lp* was quite sensitive to inhibition by neighboring *Lp* that was already growing on the plate (*Figure 4A*). Indeed, *Lp* susceptibility closely mirrored the susceptibility of *Lm* to inhibition (compare to *Figure 1A*), even though the bacterial cells secreting the inhibitor were genetically identical to the inhibited *Lp*. In both cases, we observed a sharp boundary at the edge of the zone of inhibition. In contrast, the 'no zone' *Lp* strain *hisC2*::Tn did not generate a sharp zone of inhibition against neighboring *Lp* (*Figure 4A*), suggesting that the HGA-melanin pathway was responsible for both *Lm* and *Lp* inhibition. Furthermore, we found that synthetic HGA was able to inhibit *Lp* in liquid cultures at the same concentrations that were inhibitory to *Lm* (compare *Figures 4B* and *3C*).

However, our comparisons between *Lm* and *Lp* revealed one important difference in their response to HGA inhibition. Unlike *Lm*, the *Lp* cultures exposed to HGA exhibited a population rebound after a dose-dependent growth delay, measured by both OD600 and plating for viable CFUs (*Figure 4B*, *Figure 4—figure supplement 1C*). This rebound response was shared between

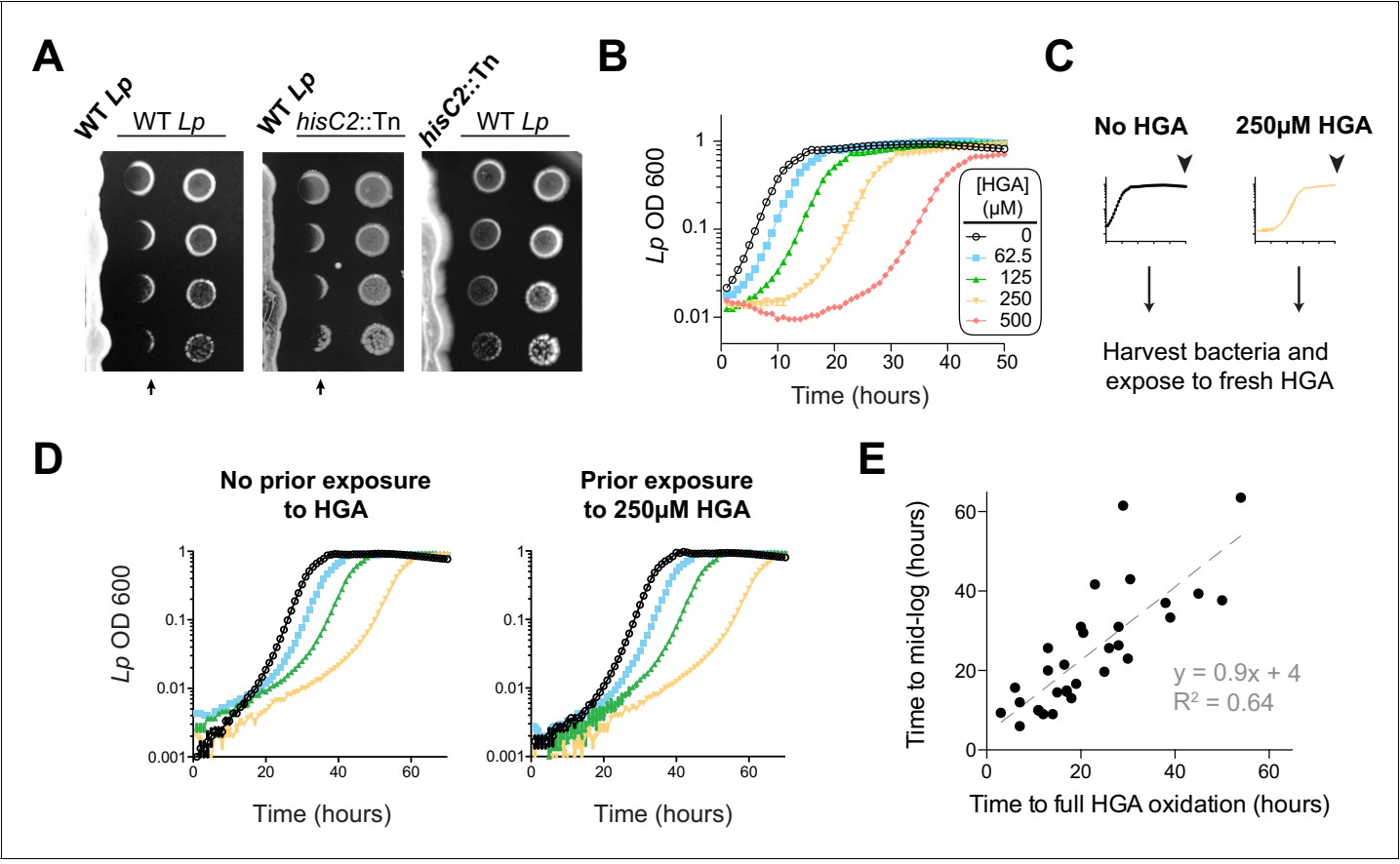

**Figure 4.** *L. pneumophila* is susceptible to bacteriostatic inhibition by HGA in rich media. (A) When pre-incubated on agar plates, *Lp* produces a zone of inhibition (arrows), preventing the growth of genetically-identical *Lp* plated 3 days later. The 'no zone' mutant *hisC2*::Tn does not produce a sharp front of inhibition, implicating HGA. (B) Increasing concentrations of synthetic HGA inhibit the growth of *Lp*, causing a growth delay in rich media. Error bars showing standard deviation are small and mostly obscured by the symbols. (C) To test if *Lp* population recovery at late time points following HGA exposure is due to the outgrowth of HGA-resistant mutants, we grew *Lp* with or without HGA and sampled bacteria at the end of the experiment (arrowhead) that were unexposed to HGA or were exposed to 250 μM HGA. These were used to inoculate media ± fresh HGA. (D) Prior HGA exposure did not lead to subsequent resistance. (E) The time for *Lp* to grow in the presence of HGA is correlated with the time for synthetic HGA to oxidize at each concentration, suggesting that *Lp* delays growth until HGA has sufficiently oxidized to lose inhibitory activity. Plot shows data combined across eight experiments. Linear fit curve and equation are shown.

DOI: https://doi.org/10.7554/eLife.46086.009

The following figure supplement is available for figure 4:

**Figure supplement 1.** HGA-induced growth delays and quantification of HGA production.
DOI: https://doi.org/10.7554/eLife.46086.010

the KS79 lab strain and the Philadelphia-1 original *Lp* strain (**Figure 4—figure supplement 1A**). We hypothesized that the *Lp* rebound response following HGA inhibition occurred because of selection and outgrowth of HGA-resistant mutants following exposure. To test this possibility, we collected 'post-rebound' stationary phase *Lp* previously exposed to 250 µM HGA and compared their subsequent HGA sensitivity to unexposed *Lp* (**Figure 4C–D**). We found that both cultures showed nearly identical susceptibility to HGA inhibition, suggesting that *Lp* population rebounds were not driven by genetic adaptation.

We therefore considered an alternate possibility that *Lp* populations exposed to HGA remain static until HGA levels fall below inhibitory concentrations, reflecting the auto-oxidation and loss of HGA activity over time. This possibility was supported by CFU measurements, which showed that HGA is bacteriostatic but not bacteriocidal against *Lp* during the growth delay (**Figure 4—figure supplement 1C**). Furthermore, we observed a strong, linear correlation between the time required to fully oxidize a given concentration of HGA and the length of the growth delay induced by *Lp* (**Figure 4E**). Based on our multiple observations, we favor the parsimonious conclusion that synthetic HGA causes initial, bacteriostatic inhibition of *Lp* until it has been sufficiently oxidized and thereby inactivated, enabling *Lp* growth. We note that the liquid culture assays (**Figure 4B**) differ from the co-plating assays, in which we did not observe *Lp* rebound (**Figure 4A**). In the latter case, we presume that bacteria continually secrete fresh HGA to replace the oxidized HGA over time. Thus, our results confirm a surprising role for HGA in both interspecies and intraspecies *Legionella* inhibition.

## Non-essential gene *lpg1681* sensitizes *L. pneumophila* to HGA

We next investigated the molecular basis of *L. pneumophila* susceptibility and resistance to HGA using bacterial genetics. First, we tested the role of the HmgA-C proteins, which break down intracellular HGA and recycle it back into central metabolism. We hypothesized that HmgA-C proteins might also be able to deactivate extracellular HGA (**Rodríguez-Rojas et al., 2009**) (**Figure 2D**). Contrary to this hypothesis, we found that the growth response of the ∆*hmgA* mutant to increasing concentrations of synthetic HGA was nearly identical to that of wild type *Lp* (**Figure 4—figure supplement 1B**). These results suggest that the intracellular recycling pathway does not play an appreciable role in *Lp* resistance to extracellular HGA.

Having excluded the obvious candidate pathway for *Lp* resistance to HGA, we pursued an unbiased forward genetics approach. Because HGA is strongly inhibitory to low density bacteria, we performed a selection for spontaneous, HGA-resistant mutants of *Lp* and *Lm* using a high HGA concentration that normally prevents almost all growth for both species. To prevent HGA from reacting with media components and becoming inactive (as in **Figure 3D and F**), we mixed the bacteria with 1 mM HGA in agar overlays poured onto low-cysteine BCYE plates (**Figure 5A**). After six days, an average of 53 colonies had grown up on each *Lp* plate under HGA selection, whereas only 3–5 colonies grew on *Lm* plates exposed to HGA. Based on these results, we focused on HGA-selected mutants of *Lp*. We retested the phenotypes of spontaneous mutants on HGA +low cysteine plates and recovered 29 *Lp* strains that consistently grew better than wild type *Lp* (**Figure 5B**). Notably, all recovered mutants had a decrease in HGA sensitivity relative to wild type, but none were completely resistant. The spontaneous mutants also showed improved growth on low-cysteine plates, relative to wild type.

To determine the underlying genetic basis of these phenotypes, we sequenced the genomes of all 29 strains plus the starting, wildtype strain of *Lp* to a median depth of 118x and identified mutations genome-wide. Each mutant strain carried 1 to 3 unique point mutations relative to the starting strain, and most of these mutations were found only in a few shared loci (**Table 1**). The most abundant category of mutants was genes related to translation; 19 of 29 resistant *Lp* strains carried mutations in translation-related machinery, of which 17 carried mutations in either elongation factor P (*lpg0287*) or enzymes responsible for adding post-translational modifications to elongation factor P (*lpg0607* and *lpg0288*). Elongation factor P acts to re-start stalled ribosomes at polyproline tracts, and its post-translational modifications are essential for these functions (**Yanagisawa et al., 2010**; **Navarre et al., 2010**; **Doerfel et al., 2013**; **Marman et al., 2014**). Based on the frequency of polyprolines in the *Lp* proteome, disruptions to elongation factor P function have the potential to impact the expression of about 33% of *Lp* proteins. The HGA resistance phenotypes we observed in these 17 *Lp* mutants could therefore result from either a large-scale shift in gene expression, or from the altered expression of specific susceptibility genes.

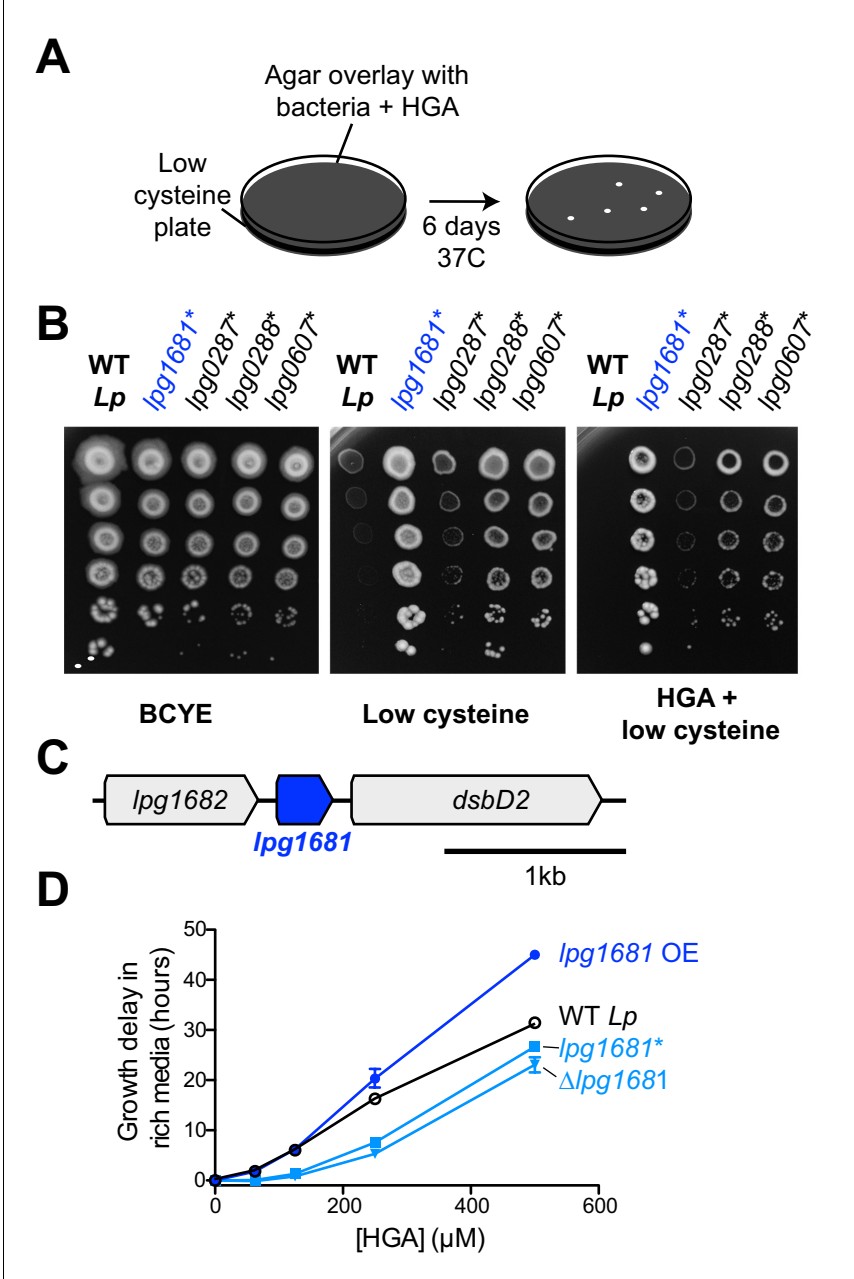

**Figure 5.** *L.pneumophila* susceptibility to HGA is modulated by *lpg1681*. (**A**) Scheme to select for *Lp* spontaneous HGA-resistant mutants. (**B**) Growth of HGA-selected mutants (*) compared to wild type *Lp*. All isolates grew better than wild type in selection conditions (HGA + low cysteine), as well as in low cysteine conditions. (**C**) Syntenic region of *lpg1681* in *Lp* strains. Lpg1681 is a hypothetical gene that lies downstream of *lpg1682*, a predicted oxidoreductase/dehydrogenase, and upstream of *dsbD2*, a thiol:disulfide interchange protein. (**D**) In rich media, a spontaneous *lpg1681* mutant (*lpg1681**) and the *lpg1681* deletion strain (Δ*lpg1681*) are less sensitive to growth inhibition by HGA than wild type *Lp*, as seen by a shorter growth delay at each concentration of HGA. Overexpression of *lpg1681* (OE) heightens sensitivity to HGA (longer growth delay). Graphs here summarize experiments similar to those in *Figure 4B*. See *Figure 5—figure supplement 2* for full data.

DOI: https://doi.org/10.7554/eLife.46086.011

The following figure supplements are available for figure 5:

**Figure supplement 1.** Evolution of genes in the *lpg1681* locus among A) *Legionella* species and B) the fish pathogen *Piscirickettsia*.

DOI: https://doi.org/10.7554/eLife.46086.012

*Figure 5 continued on next page*

*Figure 5 continued*

**Figure supplement 2.** Growth curves of *lpg1681 Lp* strains upon exposure to HGA.
DOI: https://doi.org/10.7554/eLife.46086.013

Elongation factor P disruption has also been previously linked to the activation of the stringent response pathway (*Nam et al., 2016*), which is important for coordinating a variety of stress responses in *Legionella,* including oxidative stress (*Molofsky and Swanson, 2004*; *Oliva et al., 2018*). We therefore tested if the stringent response pathway is involved in regulating HGA susceptibility or resistance. We assayed HGA susceptibility in mutant *Lp* strains with disruptions to *rpoS*, *letA*, *relA*, and *spoT*, which act early in the stringent response pathway to regulate and/or respond to the levels of the alarmone ppGpp (*Figure 7—figure supplement 1C*) (*Bachman and Swanson, 2001*; *Hammer et al., 2002*; *Dalebroux et al., 2009*). We found that the HGA susceptibility of all these mutants was similar to that of wild type; furthermore, complementation with these genes did not alter HGA susceptibility (*Figure 7—figure supplement 1D*). Thus, we find no evidence of a role for the stringent response in HGA susceptibility or resistance. Instead, we propose that disruptions to elongation factor P result in pleiotropic translation defects that together lead to HGA resistance via still-unknown mechanisms. In addition to the translation-related mutants, we found five missense mutations in *secY* or *secD* (*lpg0349* and *lpg2001*), members of the Sec secretion apparatus that moves polypeptides across the cytosolic membrane; one mutation in *aceE* pyruvate dehydrogenase (*lpg1504*); and four mutations in a hypothetical gene, *lpg1681* (*Table 1*, *Figure 5C*). The Sec apparatus is involved in the secretion of many substrates and mutations to this machinery could also lead to extensive pleiotropic defects.

Instead, we focused on the relatively uncharacterized hypothetical gene *lpg1681*, which encodes a small, 105 amino acid protein with no predicted domains apart from two transmembrane helices. This gene is adjacent in the genome to *lpg1682*, which encodes for a predicted oxidoreductase/dehydrogenase, and *lpg1680*, which encodes for the thiol:disulfide exchange protein DsbD2 (*Figure 5C*, *Figure 5—figure supplement 1*). Functional studies of DsbD2 (aka DiSulfide Bond reductase D2) have demonstrated that it interacts with thioredoxin to regulate disulfide bond remodeling in the periplasm (*Inaba, 2009*; *Kpadeh et al., 2015*). If *lpg1681* has a redox function related to its neighboring genes, we expected its syntenic locus to be conserved across bacterial strains and species. Consistent with this prediction, we found that the *lpg1680-1682* locus is present and conserved among over 500 sequenced *Lp* strains currently in NCBI databases. Outside *L. pneumophila*, *lpg1681* is mostly restricted to the *Legionella* genus, present in about half of the currently sequenced species (*Burstein et al., 2016*) (*Figure 5—figure supplement 1*). A homolog of *lpg1681* is also found in the draft genome of *Piscirickettsia litoralis*, a gamma proteobacterium and fish

**Table 1.** Genes Mutated in HGA-Selected *L. pneumophila*

| Mutated locus | Function or product | # Spontaneous Mutants w/this mutation | Mutations recovered |
|---|---|---|---|
| *lpg1681* | Hypothetical Protein | 4 | R49K, R49S, R49G, T50K |
| *lpg0288* | YjeK, 2,3-beta-lysine aminomutase | 11 | W8S, Q9*, L26G, K39*, A50P, R101L, D196G, H215R, Q243K, I250F, Q294* |
| *lpg0607* | PoxA/YjeA/GenX, Elongation factor P beta-lysine transferrase | 5 | W100*, Q184L, A218V, Q258*, 1 bp deletion in S156 |
| *lpg0325* | RpS7 ribosomal 30S protein | 1 | G100D |
| *lpg0336* | RplP 50 s ribosomal protein | 1 | G88R |
| *lpg0287* | Elongation factor P | 1 | Stop > Q (70 AA Extension) |
| *lpg0349* | SecY | 3 | N118K, Q132R, R369L |
| *lpg2001* | SecD | 2 | V238F, A277G |
| *lpg1504* | AceE pyruvate dehydrogenase | 1 | P272C |

\* =introduction of a stop codon
DOI: https://doi.org/10.7554/eLife.46086.014

pathogen (*Wan et al., 2016*). In all cases, *lpg1681* resides upstream of *dsbD2*, suggesting a functional link between these proteins (*Figure 5—figure supplement 1*) and implicating *lpg1681* in a role in redox homeostasis. We, therefore, viewed *lpg1681* as a promising candidate for a gene involved in HGA susceptibility.

We constructed *lpg1681* overexpression and deletion strains in *Lp* and tested the susceptibility of these strains to HGA. Similar to the spontaneous mutants we recovered, we found that the Δ*lpg1681* strain was more resistant to synthetic HGA in rich media (*Figure 5D*, *Figure 5—figure supplement 2*). Conversely, overexpression of *lpg1681* increased *Lp* sensitivity, resulting in longer growth delays than wild type at high concentrations of HGA. We therefore conclude that wild type *lpg1681* expression sensitizes *Lp* to inhibition by extracellular HGA. Given its genetic linkage with *DsbD*, our findings further suggest that alteration of disulfide bond regulation in the periplasm might constitute one means to mitigate HGA susceptibility.

## *L. pneumophila* susceptibility to HGA is density-dependent

Our unbiased genetic screen for *Lp* resistance to HGA only revealed mutants that were partially resistant to HGA. These mutants had a smaller growth delay than wild type at a given HGA concentration, but all remained qualitatively susceptible to inhibition. These results suggest that the genetic routes for *Lp* to completely escape from HGA-mediated inhibition are limited. Yet, *Lp* secretes abundant HGA into its local environment, despite the fact that HGA secretion is not required for *Lp* growth or metabolism (as seen by the robust growth of the *hisC2::Tn* mutants, *Figures 2* and *4*). Thus, our findings do not provide an adequate explanation for the paradox of how *Lp* cells can secrete a toxic compound to which they apparently carry no heritable resistance.

We therefore considered a distinct mechanism by which *Lp* might avoid self-inhibition: *Lp* might produce and secrete HGA only during conditions when it is not susceptible to HGA. To address this possibility, we investigated when and where *Lp* secretes HGA. We tracked HGA secretion across a growth curve of *Lp* in rich media with our previously described conditioned media assay. It has long been known that *Lp* produces abundant HGA-melanin pigment in stationary phase, when the bacteria are undergoing very slow or no growth (*Pine et al., 1979*; *Berg et al., 1985*; *Wiater et al., 1994*). By comparing to a synthetic HGA standard curve (*Figure 4—figure supplement 1F*), we estimate that *Lp* secretes a burst of HGA in stationary phase, producing between 183–266 µM of active HGA within 5 hrs (*Figure 6A*). HGA secretion then continues after the population has ceased growing. These quantities of HGA are more than enough to be inhibitory to *Lp* (*Figure 4B*).

For *Lp* to avoid self-inhibition from HGA, we hypothesized that *Lp* might be resistant to this inhibitor at high density and/or when the cells are in a stationary phase of growth. We therefore investigated if cell density or growth phase impacted HGA susceptibility (*Figure 6B*). For *Lp* exposed to HGA in rich media, we found that the growth phase of bacteria used to inoculate the experiment had little impact on HGA susceptibility (*Figure 6C*). In contrast, when cells were inoculated at high density ($10^9$/mL instead of $10^8$/mL), they were resistant even to high concentrations of HGA (*Figure 6C*), suggesting that cell density might be linked to HGA resistance. However, because both cell density and growth phase are changing over time during our assays in rich media, we could not fully separate their contributions to HGA resistance in *Lp*.

We therefore created a new assay to assess HGA susceptibility. We exposed *Lp* bacteria to HGA at different dilutions in nutrient-free PBS, which ensured that the bacteria did not grow or change cell density during the course of the experiment. After 24 hr exposure to 125 µM HGA, we assessed bacterial viability by plating for viable CFUs (*Figure 7A*). No measurable darkening of the HGA was detected in this assay, suggesting that the oxidation and de-activation of HGA was considerably slowed in low-nutrient conditions. We found that *Lp* bacteria incubated at high density with HGA (above $10^8$ CFU/mL) were largely protected from inhibition (*Figure 7A*). However, at lower density ($10^7$ CFU/mL), *Lp* bacteria were extremely sensitive to HGA, with at least a $10^6$-fold reduction in CFUs, to below our limit of detection. This result suggests that, although HGA is bacteriostatic in rich media, it appears to be strongly bacteriocidal in PBS. Nevertheless, as in the rich media assay, the resistance to HGA was dependent on cell density and was not altered by inoculum growth phase (*Figure 7A*).

The dramatic loss in viable CFUs in the PBS assay allowed us to further investigate the mechanism behind low-density susceptibility. Specifically, we asked if this loss in viability was due to HGA or one of its transient oxidative intermediates by exposing cells to HGA in both aerobic and anaerobic

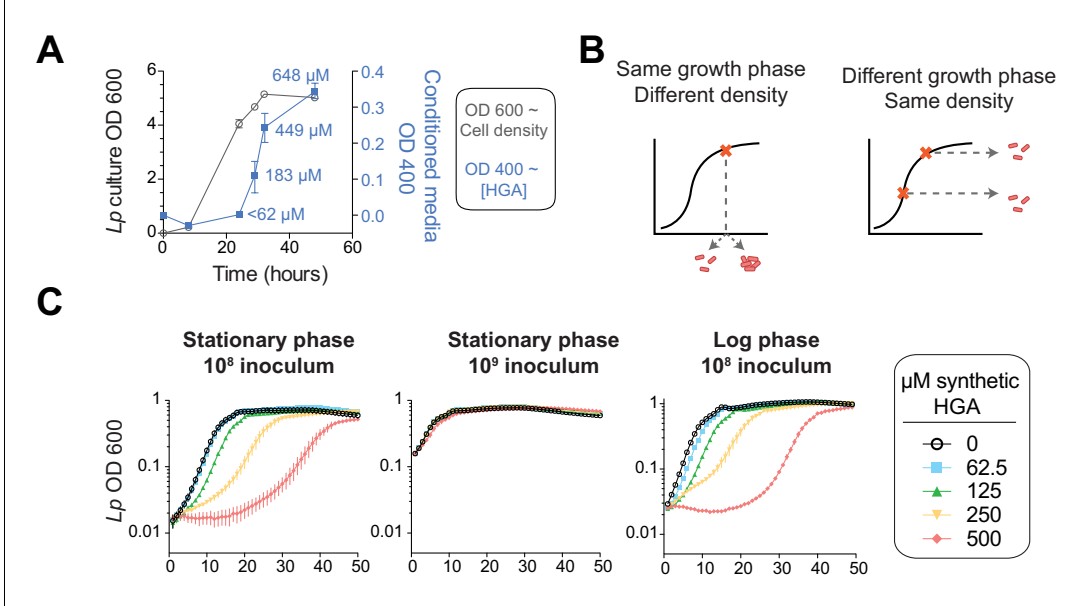

**Figure 6.** HGA susceptibility in *Lp* is linked to cell density, independent of inoculum growth phase. (**A**) Timing of HGA secretion in *Lp* in rich media, measured by OD 400 of conditioned media (CM) after allowing for full HGA oxidation (blue boxes, right y-axis). A matched growth curve of *Lp* is presented for comparison (gray circles, left y-axis). Abundant HGA is secreted during stationary phase. Estimates of secreted HGA concentration (blue) are based on a standard curve of synthetic HGA. Error bars show standard deviations. (**B**) Schematic showing how experiments controlled for inoculum density vs. growth phase. To test the impact of cell density, a single culture was diluted to multiple densities at the start of the experiment. To test the impact of growth phase, *Lp* was sampled at multiple stages of growth and diluted to the same CFU/mL at the start of the experiment. (**C**) *Lp* growth in AYE rich media is inhibited by HGA when cells are inoculated at relatively low density ($10^8$ CFU/mL), but HGA is ineffective in inhibiting growth when cells are inoculated at high density ($10^9$ CFU/mL). Growth phase of *Lp* inoculum has little impact on HGA susceptibility.

DOI: https://doi.org/10.7554/eLife.46086.015

conditions. We observed that HGA was not inhibitory to *Lp* in anaerobic conditions (*Figure 7B*), suggesting that HGA-mediated toxicity comes from the action of a reactive intermediate molecule generated during HGA oxidation.

We then considered the density-dependent difference in HGA susceptibility, reasoning that quorum sensing would be most likely to control this phenomenon. When we asked if HGA resistance depended on the previously described *Lp* quorum sensing response regulator, *lqsR* (*Tiaden et al., 2007*), we found that deleting *lqsR* had no detectable impact on HGA susceptibility or resistance (*Figure 7—figure supplement 1B*). Therefore, the density-dependent susceptibility of *Lp* to HGA must be independent of the *lqsR* pathway. We next investigated the basis of high-density resistance to HGA by *Lp* bacteria, hypothesizing that high-density cells could alter the activity of extracellular HGA, either through the secretion of inactivating compounds or through bulk, non-specific binding of HGA to bacterial biomass, leading to a reduction in its effective concentration. To test both hypotheses, we recovered the supernatants from high-density and low-density bacteria that had been incubated with or without HGA, and applied these supernatants to fresh, low-density *Lp* to assess HGA activity (*Figure 7C*). We found that the supernatants from HGA-exposed *Lp* remained fully inhibitory, even after 24 hr incubation with high-density bacteria. Furthermore, we found that adding heat-killed *Lp* bacterial cells to low-density viable *Lp* bacteria did not enhance the latter's resistance to HGA inhibition (*Figure 7—figure supplement 1A*). Therefore, we conclude that HGA susceptibility appears to be density-dependent and yet cell-intrinsic. Because *Lp* bacteria at high density both secrete and are protected from HGA, this strategy of secreting HGA only when *Lp* cells are conditionally HGA-resistant may allow *Lp* to produce a broadly active inhibitor while restricting the potential for self-harm.

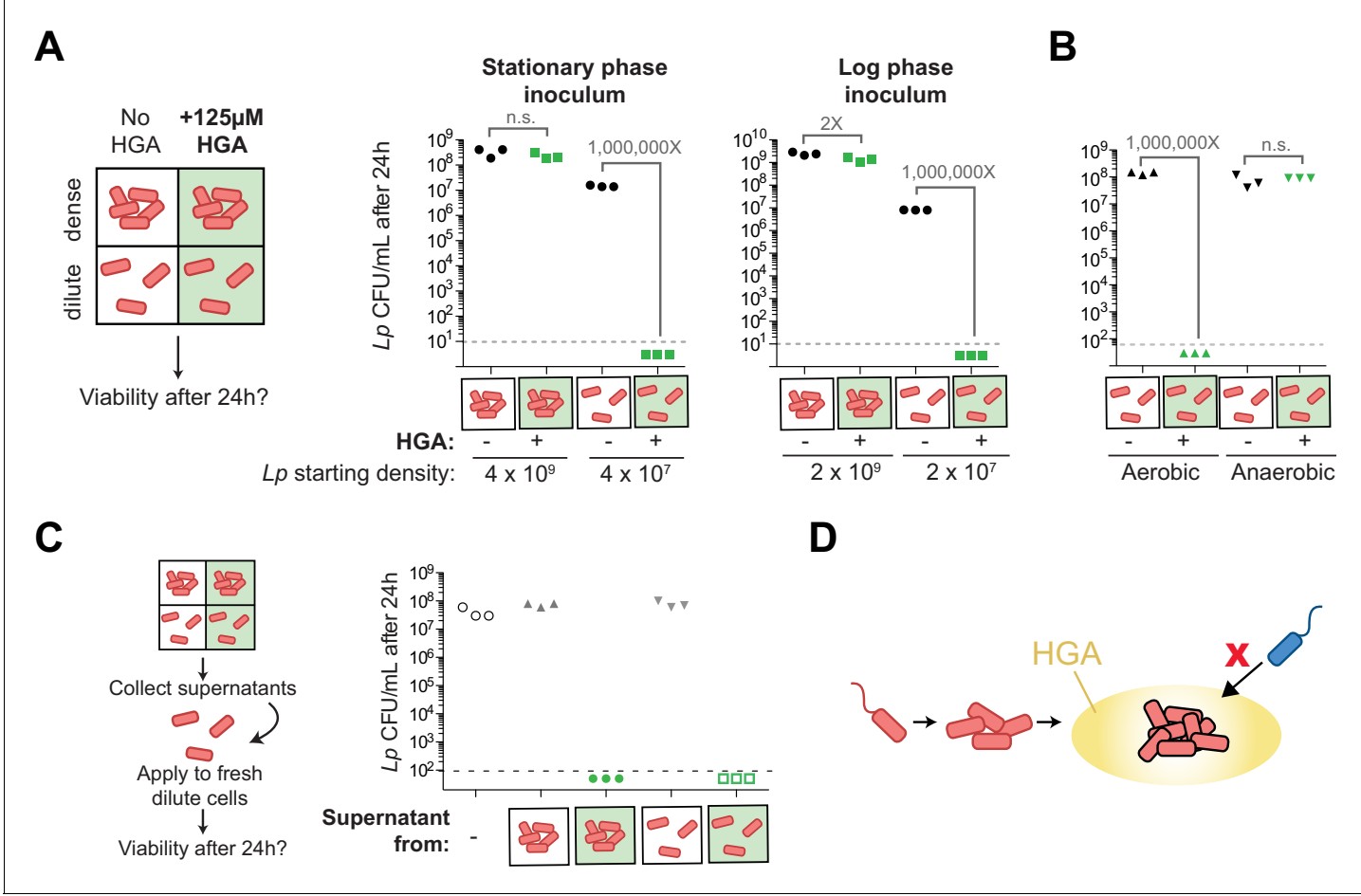

**Figure 7.** Density-dependent HGA susceptibility is heightened in low-nutrient media and requires HGA's oxidative intermediates, while high-density resistance is cell-intrinsic. (**A**) Viable CFUs following 24 hr incubation of *Lp* with or without 125 μM HGA in PBS. When incubated at high density, bacteria are almost entirely resistant to HGA, while they are highly sensitive at lower density. Dotted line shows the limit of detection. Brackets indicate the fold change in viable CFUs due to HGA exposure. (**B**) Viable CFUs following low-density HGA exposure in aerobic or anaerobic conditions. HGA is not toxic to *Lp* in the absence of oxygen. (**C**) High-density HGA-resistant cells do not inactivate extracellular HGA. Supernatants from *Lp* ±HGA at two densities were collected after 24 hr incubation and applied to new dilute cells for 24 hr. Viable CFUs were counted following supernatant exposure. HGA in all supernatants remained active against low-density *Lp*. (**D**) Model for HGA activity in extracellular environments. *Lp* (orange) colonizes a surface and grows to form a microcolony. Once cells are at high density, they secrete abundant HGA (yellow). Through unknown mechanisms, high-density *Lp* are resistant to HGA's effects (black outline), while low-density *Lp* or other *Legionella* species (blue) are inhibited by HGA and cannot invade the microcolony's niche.

DOI: https://doi.org/10.7554/eLife.46086.016

The following figure supplement is available for figure 7:

**Figure supplement 1.** Density-dependent HGA resistance acts independently of bulk cellular material, the *lqsR* quorum sensing pathway, and the stringent response pathway.

DOI: https://doi.org/10.7554/eLife.46086.017

## Discussion

The HGA-melanin pathway is well-studied and widespread among bacteria and eukaryotes. In this study, we identify HGA's oxidative intermediates as a mediator of inter-*Legionella* inhibition, both between *Legionella* species and even between genetically identical populations of *L. pneumophila*. To our knowledge, this is the first time that HGA has been described to have antimicrobial activity. One reason HGA-mediated inhibition may not have been previously documented is that the active compound(s) are redox-active, unstable molecules with transient activity. Our study finds that synthetic HGA can auto-oxidize over the course of an experiment to form inactive HGA-melanin (*Figure 3*, *Figure 4—figure supplement 1D*), allowing exposed *Lp* populations to rebound following

initial inhibition (*Figure 4B*). Intriguingly, although *Lp* populations recover upon HGA oxidation, *Lm* populations do not, suggesting that HGA may cause more harm to *Lm* cells. The quenching of HGA's inhibitory activity occurs especially rapidly in the cysteine-rich microbial media typically used to grow *Legionella* in the lab (*Figure 4—figure supplement 1D*). Conversely, we find that HGA becomes more potent in low-cysteine media or in PBS (*Figure 3F and 7*), where oxidation of HGA into HGA-melanin occurs more slowly; such nutrient-poor conditions may better replicate the nutrient-poor (oligotrophic) aquatic environments of *Legionella*'s natural habitat (*Atlas, 1999*; *Boamah et al., 2017*).

Although dense, stationary-phase cultures of *L. pneumophila* secrete abundant HGA (*Figure 6A*, *Figure 2—figure supplement 1D*), we also find that these bacteria do not possess heritable resistance to HGA (*Figure 4B–D*) and are highly susceptible to HGA-mediated inhibition at low cell density. However, they exhibit conditional, cell intrinsic resistance to HGA at high cell density (*Figure 6 and 7*). This lack of heritable resistance makes HGA-mediated inhibition different from classical antibiotics or toxins, which typically are produced by bacteria that also express resistance genes or antitoxins. HGA inhibition is also distinct from that caused by toxic metabolic by-products in two important ways. First, HGA production is not required for efficient growth or metabolism in *Legionella* species (*Figure 2—figure supplement 1D*, and see growth of *hisC2*::Tn mutant in *Figure 4A*). Second, high-density populations of *L. pneumophila* that produce HGA are themselves protected from HGA inhibition (*Figure 6 and 7*). Only neighboring, low-density *Legionella* are strongly inhibited (*Figure 4A and 6D*). The strong density-dependence of *Legionella*'s susceptibility to HGA may be another reason that it has been previously undiscovered despite intense study of these bacteria.

HGA-mediated inhibition of *Legionella* is reminiscent of the antimicrobial activities of phenazines, another class of small aromatic molecules including pyocyanin from *Pseudomonas aeruginosa*. Both types of molecules are redox-active (*Hassan and Fridovich, 1980*), are produced at high cell density (*Hassan and Fridovich, 1980*; *Baron et al., 1989*), are able to chemically react with thiol groups (*Cheluvappa et al., 2008*; *Heine et al., 2016*), and result in the production of a colored pigment (*Price-Whelan et al., 2006*). Phenazine inhibitory activity is typically thought to come from redox cycling and the production of reactive oxygen species, including $H_2O_2$ (*Hassan and Fridovich, 1980*; *Cheluvappa et al., 2008*). Oddly, HGA-melanin production has previously been implicated both in the production of (*Noorian et al., 2017*) and protection from (*Keith et al., 2007*; *Orlandi et al., 2015*) reactive oxygen species. The catalase experiments presented here have ruled out the production of extracellular $H_2O_2$ as a possible mechanism behind HGA inhibition (*Figure 3—figure supplement 1B–C*). Instead, based on association of *lpg1681* and *DsbD2* with reduced HGA sensitivity (*Figure 5*), the ability of diverse thiols to quench HGA's activity (*Figure 3—figure supplement 1E*), and precedents from phenazines (*Heine et al., 2016*), we speculate that HGA's transient, oxidative intermediates may be toxic by forming adducts on cysteine residues or otherwise disrupting disulfide bonding. Alternatively, HGA-mediated inhibition could occur via the production of other reactive oxygen species, including potentially the generation of intracellular superoxide and/or $H_2O_2$ (*Hassan and Fridovich, 1979*), which would not be affected by catalase treatment. As both of these mechanisms should be broadly inhibitory to a number of bacterial taxa, it will be interesting to survey HGA susceptibility outside of *Legionella*.

The density-dependence of *Lp*'s resistance to HGA is unusual and worthy of future study. Because high-density cells do not inactivate or bind up extracellular HGA (*Figure 7C*) and because heat-killed cells cannot protect live, low-density *Lp* from inhibition (*Figure 7—figure supplement 1A*), we infer that resistance is cell-intrinsic, resulting from differing physiology and/or gene expression between high- and low-density cells. Two pathways that commonly regulate such defenses include the stringent response pathway, which becomes active under nutrient limitation and stress, and quorum sensing pathways, which become active at high cell density (*Bachman and Swanson, 2001*; *Hammer et al., 2002*; *Tiaden et al., 2007*; *Dalebroux et al., 2009*; *Hochstrasser and Hilbi, 2017*). Although our experiments disrupting these pathways suggest that neither pathway contributes to density-dependent susceptibility or resistance (*Figure 7—figure supplement 1B–D*), we note that quorum sensing in *Legionella* remains understudied, and such pathways vary considerably across bacterial taxa (*Hochstrasser and Hilbi, 2017*; *Miller and Bassler, 2001*). Future work using unbiased approaches to investigate the regulation of HGA susceptibility may be able to uncover additional density-sensing pathways, possibly including an undescribed mode of quorum sensing in *Legionella*. Finally, we note that the density-dependent resistance was observed in well-mixed, liquid cultures in

both rich media (*Figure 6C*) and in PBS (*Figure 7A*). In natural conditions, high-density bacteria may be further protected within anaerobic regions of a biofilm, as HGA was not toxic to *L. pneumophila* in these conditions (*Figure 7B*).

L. pneumophila is often co-isolated with other *Legionella* species, which likely compete for similar ecological niches (*Wéry et al., 2008*; *Pereira et al., 2017*). HGA-mediated inter-*Legionella* inhibition therefore has a strong potential to impact the success of *Lp* in both natural and man-made environments. Because high-density, established *Lp* bacterial communities are largely resistant to HGA inhibition, these communities might use HGA to protect against low-density, invading *Legionella* competitors with little harm to themselves (*Figure 7D*). In this model, motile *Lp* can disperse, colonize a new surface, and grow into a microcolony using the locally available nutrients. In these early stages, no HGA is produced. After the *Lp* population grows up and crosses a certain cell density threshold, the cells become HGA-resistant through a cell-intrinsic mechanism. When this dense population enters stationary phase, it also begins to secrete abundant HGA into the local environment. This secreted HGA has minimal impact on the resistant, producer cells. However, it can inhibit the growth of nearby, low-density *Legionella*, whether the neighboring cells are other *Legionella* species or even genetically-identical *Lp*. Given these dynamics observed in the lab, we speculate that HGA and other such inhibitors may be deployed as a bacterial niche-protective strategy.

# Materials and methods

## Bacterial strains and growth conditions

The bacterial strains and plasmids used for this study are listed in *Supplementary file 1*. As our wild type *Legionella pneumophila (Lp)* strain, we used KS79, which was derived from JR32 and ultimately from isolate Philadelphia-1 (*de Felipe et al., 2008*; *Sadosky et al., 1993*; *Rao et al., 2013*). Compared to JR32, the KS79 strain has a *comR* deletion to enable genetic manipulation (*de Felipe et al., 2008*). We used *Legionella micdadei (Lm)* tatlock as our susceptible strain (*Garrity et al., 1980*; *Hébert et al., 1980*). Liquid cultures of *Legionella* were grown shaking in AYE rich liquid media at 37 °C (*De Jesús et al., 2013*). Unless otherwise indicated, experiments were inoculated with stationary phase *Legionella*, grown from a single colony in AYE for 16–18 hr as described, to a density of 3–4 × 10$^9$ CFU/mL. For experiments with log phase *Lp*, an overnight culture was diluted into fresh AYE at 1:8 ratio (to a density of 4–5 × 10$^8$ CFU/mL) and allowed to grow to a density of 10$^9$ CFU/mL before setting up the experiment.

To manipulate the redox state of AYE, we altered the amount of cysteine added to the media from 0.4 g/L in standard AYE to 0.1, 0.2, and 0.8 g/L. On solid media, *Legionella* were grown either on BCYE agar plates either containing the standard cysteine concentration (0.4 g/L) or 'low cysteine' (0.05 g/L) (*Feeley et al., 1979*; *Solomon and Isberg, 2000*). *E. coli* strains used for cloning were grown in LB media. Where indicated, antibiotics were used at the following concentrations in solid and liquid media, respectively; chloramphenicol (5 µg/mL and 2.5 µg/mL), kanamycin (40 µg/mL) and ampicillin (50 µg/mL and 25 µg/mL). For counter-selection steps while generating deletion strains, 5% sucrose was added to BCYE plates. For agar overlay experiments, we used 0.7% agar dissolved in water, which was kept liquid at 50 °C before pouring over low cysteine BCYE plates.

## Gene deletions and complementations

Genomic knockouts in *L. pneumophila* were generated as previously described (*Wiater et al., 1994*). Briefly, we used an allelic exchange plasmid (pLAW344) harboring chloramphenicol and ampicillin selection cassettes and the counter-selection marker SacB, which confers sensitivity to sucrose. Into this plasmid, we cloned ~ 1 kb regions upstream and downstream of the gene of interest to enable homologous recombination. Following electroporation and selection on chloramphenicol, we used colony PCR to verify insertion of the plasmid into the chromosome, before counter-selection on sucrose media. From the resulting colonies, we performed PCR and Sanger sequencing to verify clean gene deletion. For complementation, the coding region of a candidate gene was cloned into a plasmid (pMMB207c) following a ptac promoter (*Chen et al., 2004*). To induce gene expression, strains carrying pMMB207c-derived plasmids were exposed to 1 mM IPTG. All constructs were assembled using Gibson cloning (NEB Catalog #E2621).

## Inhibition assays on agar plates

To visualize inhibition between neighboring *Legionella* on solid media, a streak of approximately 5 × 10⁶ CFU of the inhibitory strain of *Lp* was plated across the center of a low cysteine BCYE plate. After 3 days growth at 37 ˚C, dilutions of susceptible *Lp* or *Lm* were plated as 10 μL spots approximately 1 cm and >2 cm from the central line. Once spots were dry, plates were then incubated for an additional 3 days before scoring for inhibition. This assay was also used to quantify the bactericidal inhibition of *Lm*, with slight modifications. Here, all *Lm* was plated in 20 μL spots at 10⁶ CFU/mL. The time of plating susceptible *Lm* was treated as t = 0. Once spots were dry, plugs were extracted from within the *Lm* spots using the narrow end of a Pasteur pipette. These plugs were transferred into media, vortexed, and plated to quantify CFU. This procedure was repeated after 48 hr at 37 ˚C to compare *Lm* viability and growth within ('near', *Figure 1B–C*) or outside ('far') of the zone of inhibition.

For inhibition assays on bacterial lawns, we plated 10 μL drops of either live *Lp* or chemical compounds on top of a lawn of 5 × 10⁷ CFU *Lm* on low cysteine BCYE, and assessed growth of the lawn after 3 days at 37 ˚C. Synthetic HGA (Sigma: #H0751) was dissolved in water at a concentration of 100 mM and filter sterilized before use. To limit the potential for HGA oxidation prior to use, 100 mM aliquots prepared in water were stored frozen at −20C and discarded after 1–2 weeks. HGA-related compounds, 2-hydroxyphenylacetic acid (Sigma: #H49804) and 3-hydroxyphenylacetic acid (Sigma: #H49901), were prepared in the same way. To test the impact of DTT (Sigma: #43819) and glutathione (oxidized: Sigma #G4376, reduced: Sigma #G6529) on HGA-mediated inhibition, filter-sterilized solutions dissolved in water were mixed in equimolar ratios with HGA, and incubated shaking at room temperature for 1 hr before spotting onto bacterial lawns. HGA-melanin pigment was prepared from *Lp* conditioned media as previously described (*Zheng et al., 2013*) from KS79, the unpigmented *hisC2*::Tn mutant, and the hyperpigmented Δ*hmgA* mutant. Briefly, conditioned media was collected and sterile filtered from 100 mL cultures of *Lp* in AYE media grown shaking at 37 ˚C for 3 days. The conditioned media was acidified to a pH of 1.5 and transferred to 4 ˚C for 2 hr to precipitate. Precipitated pigment was collected by centrifugation at 4000 x g for 15 min and then washed with 10 mM HCl. Pelleted pigment was then returned to neutral pH and resuspended in PBS at 10X before testing.

## Transposon mutagenesis screen

For random transposon insertion mutagenesis, we used a Mariner transposon from the pTO100 plasmid (*O'Connor et al., 2011*). We electroporated this plasmid into the KS79 strain and allowed cells to recover at 37 ˚C for 5 hr. To select for cells with integrated transposons, cultures were plated on BCYE/Kan/sucrose plates and incubated at 37 ˚C for 3 days before screening individual mutant colonies.

To identify transposon mutants with defects in *Lm* inhibition, we transferred each *Lp* mutant onto a low cysteine plate with a lawn of 5 × 10⁷ CFU *Lm* and visually screened for those with either small zones of inhibition or no zone of inhibition. This transfer of *Lp* mutants was achieved either by replica plating using sterile Whatman paper (Whatman: #1001150) or by manual transfer with a sterile toothpick. Plates were then incubated at 37 ˚C for 3 days and scored. All putative mutants underwent clonal re-isolation, were diluted to OD 600 of 0.1, and spotted on fresh *Lm* lawns to retest their phenotypes. To map the sites of transposon integration, we used arbitrary PCR as described in *Chen et al. (1999)*, with primers redesigned to work with the pTO100 transposon (*Supplementary file 1*). Briefly, this protocol involved two PCR steps to amplify the DNA flanking the transposon. The first step used low annealing temperatures to allow the arb1 randomized primer to bind many sites in the flanking DNA while the pTO100_F or pTO100_R primer annealed within the transposon, generating multiple products that overlapped the flanking DNA. These products were amplified in the second step PCR using the arb2 and pTO100_Rd2 primers, and we used the pTO100_Rd2 primer for Sanger sequencing. PCR programs and conditions were as in *Chen et al. (1999)*.

## HGA inhibition assays in AYE rich media

For rich media assays (e.g. *Figures 3C–F*, *4B and D*), overnight cultures of *Legionella* were diluted to 10⁸ CFU/mL in AYE, mixed with synthetic HGA (at 0, 62.5, 125, 250, or 500 μM final) or with

isolated pigment in 96 well plates, and grown shaking at 425 cpm at 37 ˚C. The cytation three imaging reader (BioTek CYT3MV) was used to monitor growth by OD 600 measurements. Because oxidized pigment from synthetic HGA is detected at OD 600 as well, each experiment included bacteria-free control wells containing media and each concentration of HGA. To correct OD 600 readings for pigment development, at each time point we subtracted the control well reading from bacterial wells that received the same concentration of synthetic HGA.

For experiments with HGA 'pre-oxidation' (*Figure 3D*), we diluted HGA in AYE media and incubated this solution shaking at 37 ˚C for 24 hr in the plate reader before adding *Lm* bacteria. Complete oxidation of HGA during the 24 hr was monitored using OD 400 to track the accumulation of HGA-melanin pigment (*Figure 4—figure supplement 1*). To test if extracellular catalase could protect from HGA inhibition (*Figure 3—figure supplement 1B*), we incubated *Lm* at 107 CFU/mL with or without 125 µM HGA and either 0, 1, 10, or 100 U/mL of bovine catalase (Sigma #C30). As a control to ensure that the catalase was active, we incubated *Lp* as above with catalase and 2 mM $H_2O_2$ (Sigma #88597).

In *Lp*, HGA inhibition in AYE rich media resulted in a growth delay, similar to an extended lag phase (*Figure 4B*). To determine if this delay was due to genetic adaptation, we sampled *Lp* after 70 hr growth with 250 µM HGA or without HGA (*Figure 4C–D*). These bacteria were washed once and resuspended in fresh AYE, before being diluted back to 108 CFU/mL and then exposed to fresh, synthetic HGA as above. To assess the correlation between HGA oxidation and the length of the *Lp* growth delay (*Figure 4E*), we pooled data from eight experiments on different days that measured wild type *Lp* (KS79) exposed to a range of HGA concentrations in AYE. We considered the 'Time to full HGA oxidation' as the length of time required for a given concentration of HGA to stop forming additional HGA-melanin, measured as the time until OD 400 readings increased by less than or equal to 0.001 units per hour. The 'Time to mid-log' was measured as the time when *Lp* exposed to that concentration of HGA had grown to an OD 600 of 0.1.

To compare sensitivity to HGA among *Lp* strains, we calculated the lag phase from the growth curve of each well using the GrowthRates program (*Hall et al., 2014*). We excluded a small number of samples where the growth curve was not well fit (R < 0.99), and then for each strain used the difference in lag time between the samples with and without HGA to calculate the growth delay due to HGA (*Figure 5D*).

## HGA inhibition assays in PBS

While we were able to manipulate inoculum growth phase and cell density in the AYE assays, during these experiments the bacteria altered their density and growth phase as they grew in rich media. To separate the impacts of cell density and growth phase on HGA susceptibility, we used a complementary assay in which we evaluated *Legionella* viability when exposed to HGA in nutrient-free PBS at different cell densities. This design ensured that the bacteria maintained a constant cell density throughout the course of the experiment. Stationary phase cultures were washed 1–2 times and resuspended in PBS. We diluted these bacteria to estimated starting concentrations of 109, 108, and 107 CFU/mL and plated for CFU at t = 0. We distributed the remaining bacteria into 96 well plates with or without 125 µM HGA. Plates were incubated shaking in a plate reader at 425 cpm at 37 ˚C for 24 hr before plating to quantify CFUs on BCYE plates. CFUs were counted after 3–4 days growth at 37 ˚C. To assess HGA toxicity in aerobic vs. anaerobic conditions, *Lp* stationary phase bacteria were prepared as above and diluted to 108 CFU/mL in PBS ± 125 µM HGA in 500 uL volumes in Celltreat bio-reaction tubes (# 229472). Aerobic samples were incubated shaking in air at 37 ˚C for 24 hr, while anaerobic samples were incubated static at 37 ˚C in an anaerobic chamber (Coy lab products, #032714) filled with 5% hydrogen, 10% CO2, and 85% N2. Samples were plated to quantify CFUs as above.

To determine if high density bacteria were protected via mass action effects that diluted out the amount of HGA per cell through binding of bulk material, we asked if the addition of dense, heat-killed bacteria could protect low-density *Lp* from HGA (*Figure 7—figure supplement 1A*). To prepare high-density heat-killed bacteria, an overnight culture of *Lp* was washed once in PBS, resuspended to 2 × 109 CFU/mL, and incubated at 100–110˚C for 60 min. After heating, this sample was diluted 1:2 and mixed with 107 CFU/mL live *Lp* in PBS ± 125 µM HGA to assess protection. As a control, 109 and 107 CFU/mL live *Lp* with or without HGA were tested simultaneously. Cells were incubated and plated as above to assess viability. To determine if high density bacteria were protected

via HGA degradation, we tested if the supernatants from HGA-exposed, high-density bacteria retained the potency to inhibit low-density bacteria (*Figure 7C*). To generate supernatants, we set up 2 mL samples containing 108 or 107 CFU/mL of *Lp* in PBS ± 125 µM HGA and incubated them shaking at 37 ˚C for 20 hr. After plating aliquots for viable CFU, we pelleted the remaining bacteria and sterile filtered 1 mL of each supernatant through a 0.2 µm filter. Each supernatant was tested in triplicate, incubated with fresh *Lp* at 107 CFU/mL in a 96 well plate as above for 24 hr before plating for CFU. As a control, 107 CFU/mL live *Lp* were incubated in PBS alone.

## Quantification of HGA's ability to react with cysteine

HGA is known to be a redox-active molecule, with a redox potential of +0.636V (*Eslami et al., 2013*). As this measurement can be altered by pH and temperature, we assessed the ability for HGA to oxidize cysteine in our experimental conditions using Ellman's reagent (also known as 5,5'-dithio-bis-(2-nitrobenzoic acid), DTNB, Invitrogen #D8451). Ellman's reagent reacts in the presence of reduced thiol groups on L-cysteine to form a yellow color, which can be read as 412 nm absorbance. When thiol groups are oxidized, the Ellman's reagent is colorless. We used the ability for HGA to decrease the amount of reduced cysteine as a proxy for its oxidizing ability. Stock solutions of both 100 mM HGA and 1.5 mM L-cysteine (Sigma #C6852) were prepared fresh in PBS at the start of the experiment. Different concentrations of HGA (from 125 µM to 8 mM) were incubated in triplicate, shaking at 25˚C with 1.5 mM cysteine in PBS for 16 hr. These conditions were compared to a standard curve of cysteine from 0 to 2 mM, which were incubated in parallel with the experimental samples to account for cysteine oxidation over time. To quantify the remaining free thiol groups, 180 uL of 0.08 mg/mL Ellman's reagent was mixed with 17.65 uL of each experimental or standard sample in a 96 well plate, incubated for 3 hr at 25˚C, and read for 412 nm absorbance. The 'decrease in reduced cysteine' was calculated as the difference between the initial and final measured cysteine concentrations, based on the standard curve conversion.

## Estimation of amount of HGA secreted by *Lp*

HGA-melanin is a black-brown pigment that is easily detected at OD 400. We took advantage of this coloration to estimate the amount of HGA that had been secreted by *Lp* by comparing the color of conditioned media to a standard curve of oxidized synthetic HGA. To isolate conditioned media from pigment mutant strains, cultures of KS79, Δ*hmgA*, and hisC2::Tn were inoculated from fresh colonies from a BCYE plate into 5 mL AYE and were grown shaking at 37 ˚C for 48 hr. We then collected conditioned media by pelleting the bacteria and passing the supernatant through a 0.2 µm filter. To harvest conditioned media for a time course, cultures of *Lp* were inoculated into 5 ml AYE and grown shaking at 37 ˚C. After 15, 20, 24, 39, 44, and 48 hr, we measured the OD 600 of the culture and collected conditioned media. To create a standard curve, we diluted synthetic HGA into AYE at the following concentrations: 62.5 µM, 125 µM, 250 µM, 500 µM, and 1 mM. The conditioned media and standard curve samples were incubated in a 96 well plate in a plate reader shaking at 37 ˚C for 24 hr to allow the HGA to oxidize. We used OD 400 data from the 24 hr time point to generate a standard curve for each HGA concentration and calculated a line of best fit using linear regression. This equation was used to estimate the amount of secreted HGA that corresponded to the OD 400 of each conditioned media sample. (*Figure 4—figure supplement 1*). In these results, we saw that the pool of HGA + HGA melanin increased by 266 µM within 5 hr. Based on the HGA-melanin production kinetics we measured in AYE (*Figure 4—figure supplement 1D*) where 250 µM HGA oxidized ~ 31% in the first 5 hr, we calculated the likely range of active HGA concentrations. If all 266 µM HGA were secreted instantaneously at the beginning of the 5 hr window, by the end of that window 266 x (1–0.31)=183 µM of active HGA would remain. Therefore, we estimate that 183–266 µM of active HGA was produced during the 5 hr window.

## HGA-resistant mutants

Because the inhibitory activity of HGA is quenched through interactions with cysteine in rich media (e.g. *Figure 3F*), it was not possible to select for HGA-resistant mutants by mixing HGA into BCYE agar. Instead, to reduce the potential for HGA to react with media components while allowing sufficient access to nutrients for mutant cells to grow, we selected for HGA-resistant mutants by mixing 4 × 107 CFU *Legionella* with 1 mM HGA in 4 mL of 0.7% molten agar and pouring this solution as

an overlay on a low cysteine BCYE plate. Plates were incubated at 37 °C for 6 days, before candidate resistant colonies were picked and clonally isolated. The HGA resistance and growth of each isolate was re-tested on overlays with or without 1 mM HGA on both regular and low cysteine BCYE.

Twenty-nine isolates were more resistant to HGA than wild type *Lp* upon retesting. We sequenced and analyzed genomic DNA from these isolates and a matched wild type strain as follows. DNA was prepared from each strain using a Purelink genomic DNA mini kit (Invitrogen, #K1820). DNA concentrations were quantified using Qubit and normalized to 0.5 ng/uL. Barcoded libraries were prepared using tagmentation according to *Baym et al., 2015Baym et al., 2015*, analyzed with QuantIT DNA quantification, pooled, and sequenced with 50 bp paired-end reads on an Illumina HiSeq 2500 machine. Reads were trimmed for quality and to remove Nextera indices with Trimmomatic (*Bolger et al., 2014*) and mapped to the Philadelphia-1 genome (*Chien et al., 2004*) using Bowtie2 with default parameters (*Langmead et al., 2009*). Coverage plots were generated for each strain using bamcoverage (*Ramírez et al., 2016*) and manually examined for evidence of large genomic deletions and amplifications. None were observed, apart from a prophage that was present in the reference genome but missing from all sequenced strains, including our wild type KS79 strain. Variants were detected for each mutant using Naive Variant Caller (*Blankenberg, 2019*). Those variants that were detected in mutant strains but not the wild type strain were considered as putative causative mutations. For each of these mutations, we inspected the mapped reads and excluded faulty variant calls that either were adjacent to coverage gaps or that did not appear to be fixed in the clonal mutant and/or wild type sequences, likely due to errors in read mapping. After this manual filtering, 1–3 well-supported mutations remained for each mutant genome. Nine of the mutants were isolated on a different day from the other mutants; in addition to various unshared mutations, these nine strains each carried exactly the same missense mutation in *rplX*, which we disregarded as a background mutation that likely arose before selection. Following this exclusion, each mutant carried only a single well-supported mutation in a coding region. Most often this coding mutation was the only mutation we detected, although one mutant carried two additional intergenic point mutations. The coding mutations were point mutations or small deletions that resulted in non-synonymous changes, frame shifts, or gene extensions. Across different mutants, the mutations we uncovered were repeatedly found in the same, few loci (*Table 1*).

## Evolution of *lpg1681*

The genes in the HGA-melanin synthesis pathway are highly conserved in diverse bacteria and across the *Legionella* genus, with all genes present in all 41 currently sequenced *Legionella spp.* genomes (*Burstein et al., 2016*). In contrast, we were able to identify *lpg1681* in only 30 *Legionella spp.* genomes, as well as a single draft genome outside the *Legionella* genus– *Piscirickettsia litoralis*, an intracellular fish pathogen (*Wan et al., 2016*). Across the *lpg1681*-containing genomes, there is evidence for extensive recombination of the flanking loci, yet *lpg1681* is always found upstream of *dsbD2*. We identified most of these homologs of *lpg1681* using a jackhmmr search (*Finn et al., 2015*), followed by cross-referencing the homologs with the *Legionella* orthology groups defined by *Burstein et al. (2016)*. From this starting set, additional *lpg1681* orthologs were identified in unannotated, intergenic regions by searching for > 200 bp open reading frames upstream of *dsbD* orthologs, and confirming the homology of these regions using MAFFT alignments (*Katoh et al., 2002*). Through this method, we located *lpg1681* in all currently sequenced *Legionella* genomes that contain an annotated *dsbD2* gene, with the exception of *L. shakespearei*. We categorized the *lpg1681*-containing loci into those with similar synteny, based on the orthology group annotations in *Burstein et al. (2016)*. We colored and provided names for the neighboring genes in *Figure 5—figure supplement 1* if they had a homolog in the *L. pneumophila* Philadelphia-1 genome that was not annotated as a hypothetical gene. To assess the conservation of the *lpg1680-lpg1682* among *L. pneumophila* strains, we used blastn in the NCBI nr and wgs databases with the full *lpg1680-lpg1682* genomic DNA sequence as a query. We found that the full region was conserved with few mutations across 501 currently sequenced *L. pneumophila* strains.

## Acknowledgements

We thank Howard Shuman for providing *Legionella* strains as well as the pLAW344 and pMMB207c plasmids, and Tamara O'Connor for the pTO100 plasmid carrying the transposon. Thank you to

Michelle Swanson for providing the stringent response pathway mutant and complementation strains. Thank you to Meera Shenoy and the Koch lab for assistance with anaerobic incubations. We thank Ben Ross, Julian Simon, Rasi Subramaniam, Howard Shuman, Pete Greenberg, Jim Imlay, and the three reviewers for useful discussion and suggestions. We also thank Michelle Hays, Kevin Forsberg, Alistair Russell, Janet Young, and Howard Shuman for providing feedback on the manuscript.

## Additional information

### Funding

| Funder | Grant reference number | Author |
|---|---|---|
| National Institute of Allergy and Infectious Diseases | 1 K99 AI139344-01 | Tera C Levin |
| Damon Runyon Cancer Research Foundation | DRG 2228-15 | Tera C Levin |
| Howard Hughes Medical Institute | | Harmit S Malik |

The funders had no role in study design, data collection and interpretation, or the decision to submit the work for publication.

### Author contributions

Tera C Levin, Conceptualization, Data curation, Formal analysis, Supervision, Funding acquisition, Validation, Investigation, Visualization, Methodology, Writing—original draft, Project administration, Writing—review and editing; Brian P Goldspiel, Data curation, Investigation, Visualization, Methodology, Writing—original draft, Writing—review and editing; Harmit S Malik, Resources, Supervision, Funding acquisition, Visualization, Project administration, Writing—review and editing

### Author ORCIDs

Tera C Levin https://orcid.org/0000-0001-7883-8522
Brian P Goldspiel https://orcid.org/0000-0001-5762-8868
Harmit S Malik http://orcid.org/0000-0001-6005-0016

### Decision letter and Author response

Decision letter https://doi.org/10.7554/eLife.46086.023
Author response https://doi.org/10.7554/eLife.46086.024

## Additional files

### Supplementary files

• Supplementary file 1. (**A**) Recovered transposon mutants of *L. pneumophila* with defects in *L. micdadei* inhibition (**B**) Strains and plasmids (**C**) Primers used for cloning.
DOI: https://doi.org/10.7554/eLife.46086.018

• Transparent reporting form
DOI: https://doi.org/10.7554/eLife.46086.019

### Data availability

All data generated or analyzed during this study are included in the manuscript and supporting files. The sequencing reads from our analyses of the HGA-selected mutants have been deposited to the Sequence Read Archive under the accession number PRJNA543158. Table 1 summarizes all of the mutations that were observed across the 29 mutant strains.

The following dataset was generated:

| Author(s) | Year | Dataset title | Dataset URL | Database and Identifier |
|---|---|---|---|---|
| Levin TC, Goldspiel | 2019 | Sequencing reads from analyses of | https://www.ncbi.nlm. | NCBI Sequence Read |

| BP, Malik HS | the HGA-selected mutants | nih.gov/bioproject/ PRJNA543158/ | Archive, PRJNA543158 |
|---|---|---|---|

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
