## [Decision Letter]

[Editors’ note: a previous version of this study was rejected after peer review, but the authors submitted for reconsideration. The first decision letter after peer review is shown below.]

Thank you for submitting your work entitled "*L. pneumophila* deploys a self-active inhibitor for inter-*Legionella* competition" for consideration by *eLife*. Your article has been reviewed by three peer reviewers, and the evaluation has been overseen by a Reviewing Editor and a Senior Editor. The following individual involved in the review of your submission has agreed to reveal their identity: Michele Swanson (Reviewer #2).

Our decision has been reached after consultation between the reviewers. Based on these discussions and the individual reviews below, we regret to inform you that your work will not be considered further for publication in *eLife* at this time.

We note that there is much to like about your manuscript. The initial description of the HGA-inhibitory activity is interesting, and the genetic experiments are well done. However, it falls short on a few fronts of the quality we seek for an *eLife* manuscript, so we are returning this version of the manuscript yet providing strong encouragement to resubmit pending revision and additional experimentation. Attention should be focused on the following things:

The design, presentation and interpretation of the cell density experiments lack the context of the well-described cellular differentiation coordinated by the stringent response for Legionella. Suggestions for how to address this are found below.

More direct experimentation to characterize the redox properties of HGA and links to ROS pathways would strengthen the mechanistic claims. Suggestions for how to address this are found below.

Ecological claims are provocative, yet overstated. Please tighten the discussion, condensing the "social goods" part yet providing more detail on evidence of interspecies Legionella competition in the environment. Use of "for inter-Legionella competition" in the title is not appropriate.

Claims of novelty for HGA having dual effects on their producers compared to phenazines are incorrect, and the discussion should be revised accordingly. Several years ago it was demonstrated that phenazines trigger lysis of the producing *Pseudomonas* population via ROS-linked effects (see papers by Das and Manefield), and a recent study describes when and how phenazines are toxic vs. beneficial to their produces (see Meirelles and Newman, 2018).

*Reviewer #1:*

While this manuscript takes an impressive genetics approach to an interesting observation, at the end of the day, the key biological insights here aren't clear. Similar things have been reported for phenazines: they can be both beneficial under some circumstances, and harmful under others, including harmful to the producing cells (they are "self-active", see work on eDNA release, for example). The more interesting question is why is HGA toxic to the producers? That, unfortunately, isn't clarified here, and without deeper insight into this aspect, this paper falls short of the *eLife* bar.

Essential revisions:

- An oxidative stress mechanism is asserted but not rigorously tested. What is the redox potential of HGA? What factors in the cell respond to it? The screen for resistance found a number of general-growth defects, but unfortunately, they don't shed much light on the mechanism.

- Insufficient physiological description of state of cultures used in experiments (cell density/growth stage of inoculum?). Energy state? Please state and measure, where possible.

- Ecological relevance seems more asserted than supported by data. Where has HGA been shown to be important?

- Discussion section is overly speculative and unfocused.

*Reviewer #2:*

Levin and colleagues identified a new activity of a pigment produced by *Legionella* – a redox-sensitive inhibition of growth. By applying a logical series of genetic and biological assays, they also identified a previously uncharacterized locus that modulates bacterial sensitivity to the HGA pigment. The authors discuss their results within a provocative ecological framework, proposing that, by secreting the pigment in stationary phase, *L. pneumophila* restricts encroachment by competitor *Legionella*.

In general, the experiments are logical, quantitative, and well controlled, and the primary conclusions are supported by data provided. The value of the manuscript is to advance the scant literature on mechanisms of ecological fitness of *L. pneumophila* in aquatic environments, the source of transmission to humans. Ambiguity concerning growth phase vs cell density complicates independent interpretation and replication of the results, a weakness that can be corrected by editing and/or comparing directly log and stationary phase WT cells in their HGA-sensitivity assay.

1) Cell density vs growth phase: Abstract, Results section, Discussion section; Figure 4, Figure 5, Figure 4—figure supplement 1 and elsewhere. *L. pneumophila* induce expression of multiple virulence and resilience traits upon entry into stationary phase, whereas exponential phase cells are more susceptible to killing by macrophages and a variety of environmental stresses (e.g. Molofsky and Swanson, 2004; Olivia et al., 2018). Accordingly, in experiments testing sensitivity of *Legionella* to HGA, clarify in Materials and methods section and relevant figure legends the growth phase of the cells that were challenged with HGA. Currently, the references to "regulatory state" (Discussion section) are vague and not tested directly here.

2) HGA induces transient "growth delay", Figure 4, Figure 5, Figure 4—figure supplement 1, Figure 5—figure supplement 2, and Results section "improvement in growth": Without knowing the growth phase of inocula used throughout this study, it is difficult to interpret the apparent prolonged lag phase of HGA-treated cells: Does HGA kill a large subset of the treated population, or instead delay entrance into the cell cycle for the entire population? If the latter, do you speculate that HGA loses activity or instead that the bacteria adapt to overcome the HGA block to replication?

3) Cell density protection, Figure 4D, Figure 4—figure supplement 1, and Results section: Do cells need to be metabolically active to confer protection from HGA, or is this a biochemical effect of distributing the oxidative damage across more substrate – more surface area of stationary phase L. pneumophila cells? For example, do UV-inactivated bacteria protect a minority population of viable stationary phase cells? Related to #1, at high densities, are log phase cells also protected from HGA treatment?

4) Title and Discussion section, "observed specificity": Whether HGA also inhibits non-*Legionella* bacteria is predicted from the Stewart et al., 2011 study of surfactant and implied here but is not tested directly. Also, this paragraph focuses on pigments, but cites Stewart et al., 2011 that only analyzes surfactant.

5) Translation mutants alter HGA susceptibility, Results section: Nam et al., 2016 reported that disruption of elongation factor P function induces the stringent response of *Salmonella*. Since the stringent response triggers entry of replicating *L. pneumophila* into stationary phase and the more resilient "transmissive" cell type (e.g. Molofsky and Swanson, 2004; Olivia et al., 2018), the authors may wish to comment on whether activation of the stringent response might account for the isolation of a large number of translation mutants in the screen for enhanced resistance to HGA.

*Reviewer #3:*

Levin et al. identify an important mechanism that *Legionella pneumophila* may employ to successfully compete in complex multispecies communities. Specifically, demonstrate that interbacterial competition between closely related *Legionella* species is mediated in part by the secretion of the inhibitory molecule homogentisic acid by *L. pneumophila*. This work is important as it reveals a novel mechanism this pathogen may use to thrive in the environment. They identify HGA as the inhibitory molecule using a random transposon mutagenesis approach and further characterize mutations that increase resistance to HGA in the producing strain. Cysteine is shown to protect cells from the toxic effects of HGA and the oxidized form of the molecule (HGA-melanin) is non-toxic, suggesting that redox activity of HGA is important for the mechanism of inhibition. The work is well done, and the manuscript is well written.

[Editors’ note: what now follows is the decision letter after the authors submitted for further consideration.]

Thank you for resubmitting your work entitled "Density-dependent resistance protects *Legionella pneumophila* from its own antimicrobial metabolite, HGA" for further consideration at *eLife*. Your revised article has been favorably evaluated by Gisela Storz (Senior Editor), a Reviewing Editor, and three reviewers.

Overall, you have done an outstanding job in responding to the first set of reviews, and this manuscript is on-track for acceptance in *eLife*. However, the reviewers made a few suggestions that are constructive, which you will likely want to incorporate into your final manuscript prior to publication. The goal here is to help you polish your work one last time, but not to add onerous new experiments. No additional new experimentation is required, but should you want to try one quick experiment independently suggested by reviewer 1 and reviewer 3, please go for it.

We look forward to receiving your final revised manuscript and processing it for publication shortly!

*Reviewer #1:*

This manuscript has improved significantly and the authors are to be commended for doing a great job in response to the previous set of reviews. The new experiments aimed at bolstering the interpretation of the HGA toxicity mechanism have elevated the work, and the comparison with phenazines in the discussion is clarifying. The manuscript is much better focused and thoughtfully discussed, and makes an interesting new contribution to the literature on secondary metabolism, an important and understudied aspect of bacterial physiology. Why much remains to be learned about the mechanism(s) underpinning the phenotypes reported here, the authors have ruled out the usual suspects and have revealed an intriguing phenomenon that will inspire future work to fill in the molecular details and test the potential ecological importance of the finding, both worthy goals.

*Reviewer #2:*

The authors have thoughtfully and rigorously addressed each of the gaps noted in their first submission. This significantly expanded and revised manuscript is impressive in its scope, incorporating a logical series of microbiological, genetic, bioinformatic, and biochemical approaches to identify a new activity for a pigment that has long been known to be produced by the environmental opportunistic pathogen *Legionella pneumophila*. Using a clever series of quantitative assays, they convincingly demonstrate that in the stationary phase, *L. pneumophila* secretes HGA which inhibits replication of other legionellae that are present at low density, whether in log or stationary phase (Figure 6F). Based on several systematic tests of multiple potential mechanisms, they demonstrate that both regulation and activity of the HGA inhibitor have novel features that warrant continued investigation. The authors discuss their data within a provocative ecological framework that will guide future research in the field.

Altogether, the study marks a significant advance for the *Legionella* field, where mechanistic knowledge of this pathogen's strategies for persistence in the environment remains scant, despite *L. pneumophila* now being the leading cause of water-associated infectious disease in the USA. The clearly written manuscript is also highly accessible for the broad audience that will be interested in this new environmental microbiology, thanks to multiple schematic diagrams that beautifully illustrate the experimental design (eg Figure 2A, Figure 3A, Figure 5A, Figure 6B, D, E).

*Reviewer #3:*

Levin et al., have modified their previous manuscript in response to earlier reviews. The focus is narrowed to highlight the role of HGA in inhibiting *Legionella* spp. and that resistance to this inhibition is density dependent the producing species, *Lp*. This inhibition is shown to be independent of secreted surfactants, disproving a hypothesis previously set forth in the field. The secreted inhibitory factor is identified as HGA using a clever genetic screen for transposon mutants, and HGA is later shown to be inhibitory at biologically relevant concentrations. They show that this inhibitory activity is due to HGA and that catalase was not protective suggesting that inhibition is not due to the production of H_2_O_2_ through side reactions. They further show evidence that HGA inhibition is bacteriostatic rather than bacteriocidal in *Lp* and perform a forward genetic selection to identify genes that allow for growth in the presence of HGA. Unsurprisingly, they identify many genes that likely have global effects, and focus on one gene that was mutated in a subset of the population. The authors speculate that periplasmic disulfide bond formation may be important to mediate the cellular response to HGA, but no direct evidence is given. This resistance response may be density, but not growth phase, dependent and cell-intrinsic.

Several new experiments strengthen this version of the manuscript and address many questions raised previously. While the authors are unable to identify a mechanism by which HGA inhibits cells or LP gains resistance, many "obvious" candidates are tested and determined to not play a role and it is likely that resistance mechanisms will have global cellular effects that are difficult to tease apart. This is an interesting first observation of the phenomenon and should be of interest to a broad group of scientists.

---

## [Author Response]

[Editors’ note: the author responses to the first round of peer review follow.]

Our decision has been reached after consultation between the reviewers. Based on these discussions and the individual reviews below, we regret to inform you that your work will not be considered further for publication in eLife at this time.We note that there is much to like about your manuscript. The initial description of the HGA-inhibitory activity is interesting and the genetic experiments are well done. However, it falls short on a few fronts of the quality we seek for an eLife manuscript, so we are returning this version of the manuscript yet providing strong encouragement to resubmit pending revision and additional experimentation. Attention should be focused on the following things:The design, presentation and interpretation of the cell density experiments lack the context of the well-described cellular differentiation coordinated by the stringent response for Legionella. Suggestions for how to address this are found below.More direct experimentation to characterize the redox properties of HGA and links to ROS pathways would strengthen the mechanistic claims. Suggestions for how to address this are found below.Ecological claims are provocative, yet overstated. Please tighten the discussion, condensing the "social goods" part yet providing more detail on evidence of interspecies Legionella competition in the environment. Use of "for inter-Legionella competition" in the title is not appropriate.Claims of novelty for HGA having dual effects on their producers compared to phenazines are incorrect, and the discussion should be revised accordingly. Several years ago it was demonstrated that phenazines trigger lysis of the producing Pseudomonas population via ROS-linked effects (see papers by Das and Manefield), and a recent study describes when and how phenazines are toxic vs. beneficial to their produces (see Meirelles and Newman, Mol. Micro.).

There were four chief criticisms and suggestions by the reviewers. They felt that our study was not as complete as desired. They provided guidance for refocusing our revision with several additional experiments, which we have now performed over the past five months. These additional experiments and their follow-ups have now resulted in an additional two and a half main figures (and some associated supplementary figures) in our revision. We have also extensively revised our manuscript to focus on our findings. We believe that our revisions and new experiments have now addressed the majority, if not all, of the reviewers’ comments. In light of the editor’s encouragement, we are therefore resubmitting our manuscript for reconsideration at *eLife*. To aid your appraisal, we highlight the main changes in our revision below. A point-by-point response to the reviewer comments follows after.

1) We have performed additional experiments to show that HGA inhibition of *L. pneumophila (Lp)* is bacteriostatic in rich media. The *Lp* population rebounds in our experiments in rich media reflect delayed growth, which occurs after HGA has oxidized and is no longer inhibitory. Specifically, we are able to rule out an alternative hypothesis of genetic adaptation i.e., that the inhibition curves reflected killing followed by outgrowth of resistant mutants. This behavior differs from nutrient-poor media like PBS, where HGA appears to be bacteriocidal.

2) We perform additional experiments in both rich media and PBS to demonstrate that *Lp* resistance to HGA is associated with cell density and not growth phase. We note that we had already ruled out the role of the known quorum sensing pathway in this resistance, concluding that this resistance must therefore constitute a distinct density dependent phenomenon.

3) We perform additional experiments to show that HGA resistance at high density appears to be cell-intrinsic, and not a result of HGA inactivation or cell mass mediated mitigation of HGA inhibition. We show that supernatants from high-density, HGA exposed cells retain activity against low-density cells, demonstrating that extracellular HGA has not been inactivated or otherwise lost activity in the high-density condition. Moreover, we show that low-density live cells, spiked into high-density heat-killed cells, are not protected from HGA. Therefore, protection at high-density is not due to generic HGA binding to excess cellular material.

4) Following up on our findings of elongation factor P mutations that increase resistance to HGA, and reviewer suggestions, we now evaluate and rule out the role of *Legionella’s* stringent response in mediating HGA resistance. We find that mutations to the stringent response pathway have no impact on either *L. pneumophila* susceptibility to HGA or on bacterial resistance at high density.

5) We have added significant experimentation beyond what was suggested by the reviewers to test the redox hypothesis for HGA inhibition. We measure the redox properties of HGA. We also test and rule out the hypothesis that HGA inhibits by generating extracellular H_2_O_2_ by testing if catalase can rescue HGA inhibition; it cannot.

6) We have minimized our previous speculation about the ecological context of this HGA inhibition and resistance, while clarifying our comparisons in the text between HGA and phenazines. This is also reflected in our rephrasing of the title as suggested by the editor to focus more on our findings and speculate less about their implications.

Reviewer #1:While this manuscript takes an impressive genetics approach to an interesting observation, at the end of the day, the key biological insights here aren't clear. Similar things have been reported for phenazines: they can be both beneficial under some circumstances, and harmful under others, including harmful to the producing cells (they are "self-active", see work on eDNA release, for example). The more interesting question is why is HGA toxic to the producers? That, unfortunately, isn't clarified here, and without deeper insight into this aspect, this paper falls short of the eLife bar.

We acknowledge that the relevant points were not clearly explained in our original manuscript, and attempt to do better in our revision.

The interesting and unexpected feature is that HGA has vastly different impacts on adjacent, genetically-identical bacteria depending on their cell density. Specifically, high-density cells produce abundant HGA, which is not harmful to the producing cells but is harmful to their low-density neighbors in the same environment. This makes the phenomenon of HGA inhibition different from the production of antimicrobials such as antibiotics or toxins, because there is not constitutive resistance via a resistance gene or antitoxin. At the same time, HGA’s mode of action is also distinct from the release of generically toxic metabolites that would harm both producing and neighboring cells (as is true for phenazines that trigger eDNA release), because *Legionella* exhibit conditional, cell intrinsic, high-density resistance (see new Figure 6C-E). This phenomenon is different from a shift in the cost-benefit balance of phenazines across environments, which we believe the reviewer is referring to here. Nevertheless, there remain parallels between the action of HGA and phenazines, which we highlight in our new Discussion section.

As mentioned in points 2, 3, and 4 above, we now present additional data in our revision investigating the mode of HGA inhibition and resistance. We show that HGA’s activity vs. *L. pneumophila* is bacteriostatic in rich media, with population rebounds occurring after HGA becomes sufficiently oxidized and inactive (Figure 3). We also demonstrate that the density-dependence of resistance is separate from growth phase and is cell-intrinsic (Figure 6).

In our new experiments investigating the basis of resistance, we have ruled out several potential mechanisms including extracellular inactivation of HGA (Figure 6E), quorum sensing via *lqsR* (Figure 7—figure supplement 1B), stringent response pathway signaling via ppGpp (Figure 7—figure supplement 1C-D), and mass action effects through excess cellular material (Figure 7—figure supplement 1A). We have also ruled out the possibility that HGA acts by generating extracellular H_2_O_2_ (Figure 3—figure supplement 1C), a mechanism that is typical for phenazines. Through these experiments, we have tested the most probable mechanisms of action and resistance for HGA. We believe that the fact that HGA does not behave according to many of these expectations (notably, expectations that were based primarily on precedents from phenazines) makes this activity even more interesting than we initially appreciated.

Essential revisions:- An oxidative stress mechanism is asserted but not rigorously tested. What is the redox potential of HGA? What factors in the cell respond to it? The screen for resistance found a number of general-growth defects, but unfortunately, they don't shed much light on the mechanism.

The reviewer raises a fair point. The redox potential of HGA has been previously measured to be +0.636V (Eslami et al., 2013); we have now added this citation to our revision. Because these values can be altered by experimental conditions such as pH, we have also now quantified the ability of HGA to reduce cysteine in our experimental conditions (Figure 3—figure supplement 1D).

Our data certainly demonstrate that only certain redox states of HGA are active (Figure 3), but they did not necessarily show that inhibition proceeds via oxidative stress. Indeed, our new finding that catalase does not protect from HGA (Figure 3—figure supplement 1B-C) shows that production of extracellular H_2_O_2_ is not involved. We have now been more precise with our wording to indicate this distinction.

In terms of our genetic screen for resistance to HGA, we do discuss two classes in detail in our revision. The first, most abundant class is that of genes involved in elongation factor P biology in translation. On the recommendation of reviewer 2, we investigate the stringent response pathway (because of its link to elongation factor P) but find it does not play a detectable role in HGA resistance. We also investigate *lpg1681*, a previously uncharacterized gene,and suggest that partial HGA resistance can result from altered regulation of periplasmic disulfide bonding. We note that all mutations we recovered in this second screen provided only partial resistance to HGA inactivation. Yet, we unexpectedly observe nearly complete phenotypic resistance to HGA when *Lp* cells are at high density. This phenotypic resistance is density dependent and cell intrinsic, but not dependent on inoculum growth phase or on activation of the stringent response or known quorum-sensing pathway. We believe it is these observations, along with our elucidation of the *Lp* population rebound in rich media, which elevate the impact of our manuscript.

- Insufficient physiological description of state of cultures used in experiments (cell density/growth stage of inoculum?). Energy state? Please state and measure, where possible.

We apologize for the scant description in our earlier version. We have now added the information to the relevant sections including in the Materials and methods section. We have also provided data in Figure 6 demonstrating that the HGA susceptibility and resistance phenotypes in the rich media and PBS assays are robust across different inoculum growth stages.

- Ecological relevance seems more asserted than supported by data. Where has HGA been shown to be important?

We agree with this criticism and have now re-focused the Discussion section to focus more on our findings and more clearly separate where we are speculating on the possible ecological implications.

- Discussion section is overly speculative and unfocused.

We agree with this criticism. We have now completely re-written the Discussion section to make it more focused in light of suggestions made by the reviewers. We have also amended the title of our paper to focus on the chief findings of our paper rather than on our speculation about its ecological implications, which we have considerably revised in the Discussion section.

Reviewer #2:Levin and colleagues identified a new activity of a pigment produced by Legionella – a redox-sensitive inhibition of growth. By applying a logical series of genetic and biological assays, they also identified a previously uncharacterized locus that modulates bacterial sensitivity to the HGA pigment. The authors discuss their results within a provocative ecological framework, proposing that, by secreting the pigment in stationary phase, L. pneumophila restricts encroachment by competitor Legionella.In general, the experiments are logical, quantitative, and well controlled, and the primary conclusions are supported by data provided. The value of the manuscript is to advance the scant literature on mechanisms of ecological fitness of L. pneumophila in aquatic environments, the source of transmission to humans.

We thank reviewer #2 for her positive comments.

Ambiguity concerning growth phase vs cell density complicates independent interpretation and replication of the results, a weakness that can be corrected by editing and/or comparing directly log and stationary phase WT cells in their HGA-sensitivity assay.

We thank the reviewer for her feedback and for the detailed suggestions. In keeping with these comments, we have now performed several additional experiments which we now summarize in Figure 6 in the revised version. These experiments decouple the growth phase from cell density, showing that resistance to HGA inhibition is both cell-intrinsic (see below), density-dependent, and independent of growth phase.

1) Cell density vs growth phase: Abstract, Results section, Discussion section; Figure 4, Figure 5, Figure 4—figure supplement 1 and elsewhere. L. pneumophila induce expression of multiple virulence and resilience traits upon entry into stationary phase, whereas exponential phase cells are more susceptible to killing by macrophages and a variety of environmental stresses (e.g. Molofsky and Swanson, 2004; Olivia et al., 2018). Accordingly, in experiments testing sensitivity of Legionella to HGA, clarify in Materials and methods section and relevant figure legends the growth phase of the cells that were challenged with HGA. Currently, the references to "regulatory state" (Discussion section) are vague and not tested directly here.

We thank the reviewer for these suggestions and citations. We have now added details about the cultures used to set up our experiments in the Materials and methods section. More importantly, we have added several new experiments in Figure 6 to show that across our assays, cell sensitivity to HGA correlates with cell density and does not depend on the growth phase of the inoculum.

2) HGA induces transient "growth delay", Figure 4, Figure 5, Figure 4—figure supplement 1, Figure 5—figure supplement 2, and Results section "improvement in growth": Without knowing the growth phase of inocula used throughout this study, it is difficult to interpret the apparent prolonged lag phase of HGA-treated cells: Does HGA kill a large subset of the treated population, or instead delay entrance into the cell cycle for the entire population? If the latter, do you speculate that HGA loses activity or instead that the bacteria adapt to overcome the HGA block to replication?

These are excellent points raised by the reviewer, which we now address in detail in the revision with several new experiments. In new Figure 4 and Figure 4—figure supplement 1, we now show multiple lines of evidence that HGA in rich media is bacteriostatic instead of bacteriocidal against *Lp*, and that the basis of recovery from the growth delay is not via genetic adaptation. Instead, multiple lines of evidence lead us to conclude that HGA auto-oxidation in rich media over time negates its inhibitory effects, and that *Lp* bacterial cells are able to recover and resume growth after HGA concentrations drop below inhibitory levels.

3 Cell density protection, Figure 4D, Figure 4—figure supplement 1, and Results section: Do cells need to be metabolically active to confer protection from HGA, or is this a biochemical effect of distributing the oxidative damage across more substrate – more surface area of stationary phase L. pneumophila cells? For example, do UV-inactivated bacteria protect a minority population of viable stationary phase cells? Related to #1, at high densities, are log phase cells also protected from HGA treatment?

We thank the reviewer for these thoughtful suggestions, which we have gratefully used to motivate and perform new experiments. First, we show that in the PBS assay, both log and stationary phase cells are protected from HGA at high densities, but susceptible at low densities (new Figure 6C). We also find that high-density, heat-killed bacteria do not protect low-density live bacteria from HGA, suggesting that protection is not merely from abundant surface area or substrate (Figure 7—figure supplement 1A). Finally, we find that the conditioned media from high-density cells exposed to HGA retains the activity to kill low-density cells, showing that the HGA is not inactivated or adsorbed out of solution in high-density conditions (Figure 6E). Based on these experiments, we conclude that high-density protection is a cell-intrinsic effect, rather than a biochemical one of HGA dilution or distribution across more cells.

4) Title and Discussion section, "observed specificity": Whether HGA also inhibits non-Legionella bacteria is predicted from the Stewart et al., 2011 study of surfactant and implied here but is not tested directly. Also, this paragraph focuses on pigments, but cites Stewart et al., 2011 that only analyzes surfactant.

Yes, we agree with this point. We have rephrased these sections to be more precise, and removed this section from the revised, refocused Discussion section.

5) Translation mutants alter HGA susceptibility, Results section: Nam et al., 2016 reported that disruption of elongation factor P function induces the stringent response of Salmonella. Since the stringent response triggers entry of replicating L. pneumophila into stationary phase and the more resilient "transmissive" cell type (eg Molofsky and Swanson, 2004; Olivia et al., 2018), the authors may wish to comment on whether activation of the stringent response might account for the isolation of a large number of translation mutants in the screen for enhanced resistance to HGA.

We agree with the reviewer’s suggestion. We had not fully explored the potential link between elongation factor P and *Legionella’s* stringent response in our previous manuscript. We now discuss the stringent response in light of our resistant mutant screen. We also perform experiments with *Lp* bacteria strains with mutations in the stringent response pathway. However, we find that these mutations do not appear to alter HGA susceptibility or high-density resistance (Figure 7—figure supplement 1C-D). In the discussion, we note that the resistant mutants we have recovered in our screen show only partial resistance to HGA (Figure 5). Therefore, it is possible that high-density cells are able to fully resist HGA via another mechanism.

Reviewer #3:Levin et al. identify an important mechanism that Legionella pneumophila may employ to successfully compete in complex multispecies communities. Specifically, demonstrate that interbacterial competition between closely related Legionella species is mediated in part by the secretion of the inhibitory molecule homogentisic acid by L. pneumophila. This work is important as it reveals a novel mechanism this pathogen may use to thrive in the environment. They identify HGA as the inhibitory molecule using a random transposon mutagenesis approach and further characterize mutations that increase resistance to HGA in the producing strain. Cysteine is shown to protect cells from the toxic effects of HGA and the oxidized form of the molecule (HGA-melanin) is non-toxic, suggesting that redox activity of HGA is important for the mechanism of inhibition. The work is well done, and the manuscript is well written.

We thank the reviewer for their generous comments!

[Editors' note: the author responses to the re-review follow.]

Overall, you have done an outstanding job in responding to the first set of reviews, and this manuscript is on-track for acceptance in eLife. However, the reviewers made a few suggestions that are constructive, which you will likely want to incorporate into your final manuscript prior to publication. The goal here is to help you polish your work one last time, but not to add onerous new experiments. No additional new experimentation is required, but should you want to try one quick experiment independently suggested by reviewer 1 and reviewer 3, please go for it.We look forward to receiving your final revised manuscript and processing it for publication shortly!

Thank you very much! We have elected to perform this experiment and discuss its new results below.

Reviewer #1:[…] The experiments with cysteine and transposon mutagenesis point toward disruption of disulfide bonds and/or cysteine reactivity as plausible mechanisms of toxicity (even if not the only mechanisms). One straightforward experiment that might be helpful in testing whether oxidative intermediates are responsible for HGA toxicity would be for the authors to expose the cells to HGA under anoxic conditions, followed by a wash and plating under oxic conditions (in comparison to exposure and plating under oxic conditions).

Thank you to both reviewer 1 and reviewer 3 for the great suggestion. We have now performed this experiment, exposing *L. pneumophila* to HGA in both aerobic and anaerobic conditions. Instead of doing a wash, we immediately diluted the samples to plate CFUs in rich media, which we reasoned would dilute and/or inactivate the HGA below an active concentration. We found that HGA had little to no toxicity under anaerobic conditions, suggesting that the toxicity is indeed due to some of HGA’s transient oxidative intermediates. The results are now included in Figure 7B, and discussed briefly in the text.

Reviewer #3:Levin et al., have modified their previous manuscript in response to earlier reviews. The focus is narrowed to highlight the role of HGA in inhibiting Legionella spp. and that resistance to this inhibition is density dependent the producing species, Lp. This inhibition is shown to be independent of secreted surfactants, disproving a hypothesis previously set forth in the field. The secreted inhibitory factor is identified as HGA using a clever genetic screen for transposon mutants, and HGA is later shown to be inhibitory at biologically relevant concentrations. They show that this inhibitory activity is due to HGA and that catalase was not protective suggesting that inhibition is not due to the production of H2O2 through side reactions. They further show evidence that HGA inhibition is bacteriostatic rather than bacteriocidal in Lp and perform a forward genetic selection to identify genes that allow for growth in the presence of HGA. Unsurprisingly, they identify many genes that likely have global effects, and focus on one gene that was mutated in a subset of the population. The authors speculate that periplasmic disulfide bond formation may be important to mediate the cellular response to HGA, but no direct evidence is given. This resistance response may be density, but not growth phase, dependent and cell-intrinsic.

We appreciate the reviewer’s summary and appreciation for the work. We also agree that we have not definitively proved that HGA toxicity occurs via periplasmic disulfide bond disruption. This speculation was intended to help motivate future, follow-up work. We have modified the language in the Discussion section to make this distinction more clear.

Several new experiments strengthen this version of the manuscript and address many questions raised previously. While the authors are unable to identify a mechanism by which HGA inhibits cells or LP gains resistance, many "obvious" candidates are tested and determined to not play a role and it is likely that resistance mechanisms will have global cellular effects that are difficult to tease apart. This is an interesting first observation of the phenomenon and should be of interest to a broad group of scientists.

Thank you, we agree that the phenomenon is interesting and striking.